# *MetaAligner*: Towards Generalizable Multi-Objective Alignment of Language Models

**Kailai Yang[1]   Zhiwei Liu[1]   Qianqian Xie[2]\*   Jimin Huang[2]**
**Tianlin Zhang[1]   Sophia Ananiadou[1]**
[1] The University of Manchester    [2] The Fin AI
{kailai.yang,zhiwei.liu,sophia.ananiadou}@manchester.ac.uk
{xqq.sincere,zhangtianlin668}@gmail.com;jimin@chancefocus.com

## Abstract

Recent advancements in large language models (LLMs) focus on aligning to heterogeneous human expectations and values via multi-objective preference alignment. However, existing methods are dependent on the policy model parameters, which require high-cost repetition of their alignment algorithms for each new policy model, and they cannot expand to unseen objectives due to their static alignment objectives. In this work, we propose *Meta-Objective Aligner* (*MetaAligner*), the first policy-agnostic and generalizable method for multi-objective preference alignment. *MetaAligner* models multi-objective alignment into three stages: (1) **dynamic objectives reformulation** algorithm reorganizes traditional alignment datasets to supervise the model on performing flexible alignment across different objectives; (2) **conditional weak-to-strong correction** paradigm aligns the weak outputs of fixed policy models to approach strong outputs with higher preferences in the corresponding alignment objectives, enabling plug-and-play inferences on any policy models, which significantly reduces training costs and facilitates alignment on close-source policy models; (3) **generalizable inference** method flexibly adjusts target objectives by updating their text descriptions in the prompts, facilitating generalizable alignment to unseen objectives. Experimental results show that *MetaAligner* achieves significant and balanced improvements in multi-objective alignments on 10 state-of-the-art policy models, and saves up to 93.63% of GPU training hours compared to previous alignment methods. The model also effectively aligns unseen objectives, marking the first step towards generalizable multi-objective preference alignment. This project is open-sourced here.

## 1   Introduction

The recent advancements in large language models (LLMs) have focused on generating high-quality responses that align with human expectations and values. At the final stage of alignment, LLMs are supervised on human preference data via reinforcement learning from human feedback (RLHF) [40, 22, 27], where a proxy, directly trained on human preferences data, is leveraged to provide scalar rewards for reinforcement learning (RL) on the target model [22].

However, human expectations and values include a broad spectrum of heterogeneous and multi-dimensional objectives, which makes scalar supervisions inefficient for aligning diverse and inclusive human preferences [3, 24]. These drawbacks motivate further exploration into multi-objective alignment algorithms. Some intuitive methods extend RLHF into multi-objective RLHF (MORLHF) [26, 19, 24]. Due to its substantial computational cost [19, 24] and the unstable nature

---

\*   Corresponding author.

38th Conference on Neural Information Processing Systems (NeurIPS 2024).

Table 1: Comparisons between previous alignment methods and *MetaAligner* on different features. "Policy-Agnostic Alignment" means the alignment algorithm is independent of the target policy model parameters, and "Generalizability" denotes zero-shot alignment capability on unseen objectives.

| Algorithm | Paradigm | Multi-Objective Alignment | Policy-Agnostic Alignment | Generalizability |
|---|---|---|---|---|
| RLHF [22] | PPO | ✗ | ✗ | ✗ |
| MORLHF [19] | PPO | ✔ | ✗ | ✗ |
| MODPO [10, 39] | SFT, DPO | ✔ | ✗ | ✗ |
| RiC [35] | SFT | ✔ | ✗ | ✗ |
| *Aligner* [12] | SFT | ✗ | ✔ | ✗ |
| *MetaAligner* | SFT | ✔ | ✔ | ✔ |

of the proximal policy optimization (PPO) [25, 15, 23] algorithm, other methods seek to bypass the RL paradigm with multi-objective direct preference optimization (MODPO) [39, 10] or supervised fine-tuning (SFT)-based methods [35, 10], which customized prompting strategies to incorporate multiple reward values into queries explicitly.

The above methods for multi-objective alignment bear one commonality: the dependence on the policy model's parameters. This paradigm inevitably brings two key limitations: (1) they require repetition of their high-cost alignment algorithms for each newly-introduced policy model, which is incompatible with the increasing sizes and fast iteration of current foundation models [1, 30, 6, 29]; (2) all target models are statically aligned on pre-determined (e.g. "Helpful", "Harmless", "Honest" [39, 10]) objectives, with currently no efforts in expanding and evaluating their capabilities on unseen objectives. This ignorance leads to poor generalizability of existing multi-objective alignment methods.

In this work, we propose *Meta-Objective Aligner* (*MetaAligner*), the first policy-agnostic and generalizable method for multi-objective preference alignment. *MetaAligner* models multi-objective alignment into three stages: (1) the **dynamic objectives reformulation** algorithm reorganizes traditional alignment datasets into dynamic-objective alignment datasets, training *MetaAligner* to perform flexible alignment across different objectives. It achieves this by incorporating and combining text descriptions of various alignment objectives in a prompt-based manner; (2) the **conditional weak-to-strong correction** paradigm aligns the weak outputs of policy models to approach strong outputs with higher preferences in the corresponding alignment objectives. During training, *MetaAligner* is stacked onto policy models to perform objective-aware corrections, where parameters of the policy model are fixed and *MetaAligner* is optimized with an SFT-based three-step training process: warming up, equal-preference alignment, and contrastive-preference alignment. This paradigm enables *MetaAligner* to perform plug-and-play inferences on any policy models even without access to their parameters, which significantly reduces training costs and facilitates alignment on close-source LLMs; (3) the **generalizable inference** method flexibly adjusts target objectives by updating their text descriptions in the prompts. This method can also adapt *MetaAligner* to unseen objectives and achieve new alignment strategies via in-context learning [14], a new feature with rare previous exploration in alignment of language models. The number of aligned objectives also becomes expandable, theoretically leading to unlimited simultaneous alignment objectives. Table 1 compares key features between *MetaAligner* and previous methods. As shown, conditional weak-to-strong correction of *MetaAligner* extends *Aligner* [12] to multi-objective alignment scenarios, which are not directly solvable by *Aligner* itself. *MetaAligner* is also the first multi-objective alignment method to achieve policy-agnostic alignment and generalization to unseen objectives, two key advantages over previous methods such as MORLHF, MODPO, and SFT-based methods.

In summary, our main contributions are: (1) we propose *MetaAligner*, the first policy-agnostic method for multi-objective preference alignment. It performs multi-objective alignment efficiently, without tuning the policy models or accessing their parameters. Experimental results show that *MetaAligner* outperforms previous alignment methods and saves up to 93.63% of GPU training hours; (2) we utilize *MetaAligner* to exert zero-shot preference alignment for unseen objectives. To our knowledge, this work marks the first attempt at generalizable multi-objective preference alignment. Experimental results show that *MetaAligner* can simultaneously perform effective alignment for six unseen objectives while maintaining performance on aligned objectives; (3) We examine *MetaAligner* on three preference alignment datasets. Experimental results show that *MetaAligner* improves

win rates on multiple objectives across 10 policy models, substantially enhancing responses of state-of-the-art foundation models such as GPT-3.5-Turbo [21] and Claude-3 [2].

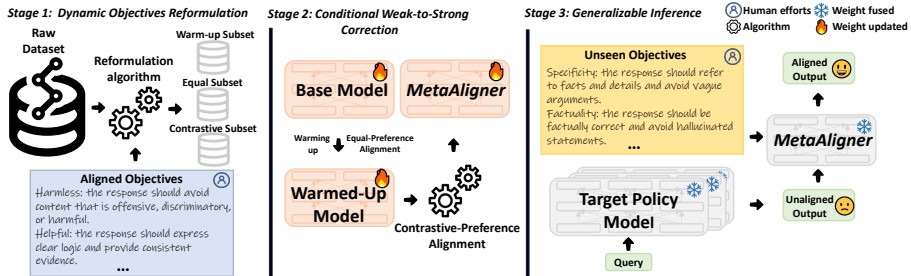

Figure 1: Illustrations of *Meta-Objective Aligner*, which follows a three-stage paradigm.

## 2 Multi-Objective Alignment of Language Models

In real-world scenarios, human expectations of high-quality responses from AI agents involve considerable variability, with complex interplays such as contradiction (e.g. "Helpful" and "Harmless") [10, 35] and dependence (e.g. "Correct" and "Informative") [34]. Multi-objective preference alignment tackles this challenge by aiming to optimize multiple objectives simultaneously. For each query-response pair, the reward vector is formalized as: $\mathbf{R}(q, y) = [r_1(q, y), ..., r_N(q, y)]^{\mathrm{T}}$, where $q, y$ denote a query and a corresponding response, $r_i$ denotes the reward values for $i$-th objective, which is defined, in most cases of preference alignment, under the Bradley-Terry [5] model of preferences. Specifically, for the same prompt $q$ and two responses $(y_1, y_2)$ under data distribution $\mathcal{D}$, the model assumes:

$$P_{\mathcal{D}}(y_1 \succ y_2 | q, i) = \sigma(r_i(q, y_1) - r_i(q, y_2)) \tag{1}$$

where $\sigma$ denotes the logistic function and $P_{\mathcal{D}}(y_1 \succ y_2)$ denotes the probability that $y_1$ is preferred against $y_2$. MORLHF aims to achieve Pareto optimal among objectives, where the policy model is optimized to maximize a linear scalarization of multiple rewards [26, 19] with a KL-divergence regularization:

$$\underset{\pi_\phi}{argmax} \, \mathbb{E}_{q \sim \mathcal{D}, y \sim \pi_\phi(y|q)} \left[ \omega^{\mathrm{T}} \mathbf{R}(q, y) \right] - \beta \mathbb{D}_{KL} \left[ \pi_\phi(y|q) \| \pi_{ref}(y|q) \right] \tag{2}$$

where $\pi_\phi$ denotes the aligned policy model parameterized by $\phi$, $\pi_{ref}$ denotes the reference policy model, $\omega = [\omega_1, ..., \omega_N] \, s.t. \sum_{i=1}^{N} \omega_i = 1, \omega_i \geq 0$ is the pre-determined heuristic target preference vector. Another paradigm directly built alignment between multiple reward values and their corresponding response by minimizing an SFT loss for the policy model:

$$\underset{\pi_\phi}{argmin} -\mathbb{E}_{(q,y) \sim \mathcal{D}} \left[ log \, \pi_\phi(y|q, \mathbf{R}(q, y)) \right] \tag{3}$$

where objectives and their corresponding reward values are described with text markers and combined into queries with a static prompting template. Compared to MORLHF, SFT-based multi-objective alignment is proven more cost-efficient and training-stable [35, 10].

## 3 Meta-Objective Aligner

Existing methods for multi-objective alignment generally face challenges in increasing training costs with new policy models and generalization to unseen objectives. To tackle these challenges, we introduce *MetaAligner*, which follows a three-stage paradigm: (1) dynamic objectives reformulation for building dynamic multi-objective datasets; (2) conditional weak-to-strong correction for model training; (3) generalizable inference for multi-objective alignment. The paradigm is illustrated in Figure 1.

### 3.1 Dynamic Objectives Reformulation

We propose a dynamic objectives reformulation algorithm to construct a dynamic multi-objective dataset, which triggers *MetaAligner*'s ability for flexible adjustment of alignment objectives. Specifically, any typical multi-objective preference alignment dataset $\mathcal{D}_m$ with $m$ samples and $N$ objectives

can be re-organized as $\{q_i, y_{i1}, y_{i2}, P_i\}_{i=1}^m$, where $P_i = [p_{i1}, ..., p_{iN}]^T$ and $p_{ij} \in \{\succ, \prec, \equiv\}$ indicates the preference on $j$-th objective.

---

**Algorithm 1** Dynamic objectives reformulation.

---

**Require:** Raw dataset $D_m : \{q_i, y_{i1}, y_{i2}, P_i\}_{i=1}^m$;
     Objective text descriptions: $[\langle d_1 \rangle, ..., \langle d_N \rangle]$;
     Prompting template: $\mathcal{T}(q, y, \mathcal{O}, t)$
**Ensure:** Contrastive subset $\mathcal{D}_c$; Equal subset $\mathcal{D}_e$.
1:   $\mathcal{D}_c \leftarrow \varnothing, \mathcal{D}_e \leftarrow \varnothing$   ▷ Initialize the 2 subsets.
2:   **for** $i \in \{1, ..., m\}$ **do**    ▷ Loop on instances.
3:      $\mathcal{O}_\succ \leftarrow \varnothing, \mathcal{O}_\prec \leftarrow \varnothing, \mathcal{O}_\equiv \leftarrow \varnothing$
4:      **for** $j \in \{1, ..., N\}$ **do**
5:         **if** $p_{ij}$ is $\succ$ **then**        ▷ Collect the objectives where $y_{i1}$ outperforms $y_{i2}$.
6:           $\mathcal{O}_\succ \leftarrow \mathcal{O}_\succ \cup \{\langle d_j \rangle\}$
7:         **else if** $p_{ij}$ is $\prec$ **then**      ▷ Collect the objectives where $y_{i2}$ outperforms $y_{i1}$.
8:           $\mathcal{O}_\prec \leftarrow \mathcal{O}_\prec \cup \{\langle d_j \rangle\}$
9:         **else** ▷ Collect the objectives where $y_1$ and $y_2$ performs equally.
10:          $\mathcal{O}_\equiv \leftarrow \mathcal{O}_\equiv \cup \{\langle d_j \rangle\}$
11:         **end if**
12:      **end for**
13:      **if** $\mathcal{O}_\succ \neq \varnothing$ **then** ▷ Build the training pairs where $y_{i1}$ is used as the target.
14:         $t \leftarrow better$
15:         $\mathcal{O}_\succ \leftarrow random\_shuffle(\mathcal{O}_\succ)$
16:         $\mathcal{D}_c \leftarrow \mathcal{D}_c \cup \{(\mathcal{T}(q_i, y_{i2}, \mathcal{O}_\succ, t), y_{i1})\}$
17:      **end if**
18:      **if** $\mathcal{O}_\prec \neq \varnothing$ **then** ▷ Build the training pairs where $y_{i2}$ is used as the target.
19:         $t \leftarrow better$
20:         $\mathcal{O}_\prec \leftarrow random\_shuffle(\mathcal{O}_\prec)$
21:         $\mathcal{D}_c \leftarrow \mathcal{D}_c \cup \{(\mathcal{T}(q_i, y_{i1}, \mathcal{O}_\prec, t), y_{i2})\}$
22:      **end if**
23:      **if** $\mathcal{O}_\equiv \neq \varnothing$ **then** ▷ Build equally-preferred training pairs.
24:         $t \leftarrow equal$
25:         $\mathcal{O}_\equiv \leftarrow random\_shuffle(\mathcal{O}_\equiv)$
26:         $\mathcal{D}_e \leftarrow \mathcal{D}_e \cup \{(\mathcal{T}(q_i, y_{i2}, \mathcal{O}_\equiv, t), y_{i1})\}$
27:      **end if**
28: **end for**

---

We define a text description for each objective: $[\langle d_1 \rangle, ..., \langle d_N \rangle]$, where $\langle d_j \rangle$ denotes the natural language description for $j$-th objective. Some examples of such descriptions are in Figure 1 and a full list is in Appendix C. With a pre-defined prompting template $\mathcal{T}$, we build a contrastive subset $\mathcal{D}_c$ and another equal subset $\mathcal{D}_e$ from $\mathcal{D}_m$. $\mathcal{D}_c$ includes all contrastive response pairs where $p \in \{\succ, \prec\}$ and $\mathcal{D}_e$ includes all equal response pairs where $p \in \{\equiv\}$. For example, we utilize the following template $\mathcal{T}(q, y, \mathcal{O}, t)$ in building for the IMHI [34] dataset:

> $[\mathcal{T}(q, y, \mathcal{O}, t)]$ Edit the following Question-Answer pair to make it $\{t\}$ considering the following objectives $\{\mathcal{O}\}$ | Question: $\{q\}$ | Answer: $\{y\}$ | Edit:

where $q$ denotes the query, $y$ denotes a corresponding response, $\mathcal{O}$ denotes the concatenation of text descriptions for the target objectives, and $t \in \{equal, better\}$ depends on the current building subset. Details of the dynamic objectives reformulation algorithm are described in Algorithm 1. For an instance within the processing loop (line 2): $\{q, y_1, y_2, P\}$, the algorithm performs a two-step reformulation: (1) collect the sets $\mathcal{O}_\succ, \mathcal{O}_\prec, \mathcal{O}_\equiv$ that includes objectives where $y_1$ outperforms $y_2$, $y_2$ outperforms $y_1$, and both perform equally (lines 3-12); (2) for each objective set $\mathcal{O}$, we randomly shuffle the objectives to further trigger the model's flexible alignment ability, and build query-response pairs based on the corresponding prompting template $\mathcal{T}$ (line 13-27). All prompting templates and examples of the algorithm are presented in Appendix C.

Training on dynamic multi-objective datasets provides three key advantages: (1) instance-level alternation of the target objectives during training enables *MetaAligner* to perform flexible alignment under different combinations of objectives; (2) mutual alignment between the same response pairs on different objectives fully leverages the supervision information in the preference vectors. (3) the reward-free alignment method (no explicit preference values required) avoids complicated preference-to-reward mapping [35] process in previous SFT-based multi-objective alignment methods.

## 3.2 Conditional Weak-to-Strong Correction

Based on the dynamic multi-objective training datasets, we train *MetaAligner* in a conditional weak-to-strong correction manner, which follows an SFT-based training objective and a three-step training paradigm.

### 3.2.1 Training Objective Derivation

*MetaAligner* is a standard conditional seq-to-seq model on top of the original policy model $\pi_\phi$, which re-distributes the policy model output $y_0$ considering objectives $\mathcal{O}$ as follows:

$$\pi^*(y|q) = \delta_\theta(y|\mathcal{T}(q, y_0, \mathcal{O}, t))\pi_\phi(y_0|q) \tag{4}$$

where $\delta_\theta$ denotes the *MetaAligner* module parameterized by $\theta$, $t$ depends on the training dataset. Conditional weak-to-strong correction directly trains *MetaAligner* to align the weak policy model output $y_0$ to the strong target output $y$, which has higher preference values in corresponding objectives $\mathcal{O}$. We have the standard cross-entropy loss as the training objective:

$$\underset{\theta,\phi}{argmin}\, \mathcal{L}(\theta,\phi;\mathcal{D}) = -\mathbb{E}_{(q,y,\mathcal{O})\sim\mathcal{D}}\left[log\,\pi^*(y|q)\right]$$

$$= -\mathbb{E}_{(q,y,\mathcal{O})\sim\mathcal{D}}\left[log\,\delta_\theta(y|\mathcal{T}(q,y_0,\mathcal{O},t))\right] - \mathbb{E}_{q\sim\mathcal{D}}\left[log\,\pi_\phi(y_0|q)\right] \quad (5)$$

We fix the parameters of the policy model, thus excluding $\phi$ from the weight update process. In practice, we use the dynamic multi-objective dataset for supervision, where the weak response in each query-response pair is directly leveraged as samples $y_0$ from unknown policy models. Therefore, we eliminate the second term in Eqn. 5 and simplify the training objective as:

$$\underset{\theta}{argmin}\, -\mathbb{E}_{(q,y_0,y,\mathcal{O})\sim\mathcal{D}}\left[log\,\delta_\theta(y|\mathcal{T}(q,y_0,\mathcal{O},t))\right] \quad (6)$$

The above action poses two advantages: (1) the computation resources required for *MetaAligner* training is detached from policy model size, which enables policy-agnostic and cost-efficient alignment for large policy models; (2) *MetaAligner* works only via outputs from the policy models, which allows training and inference for alignment on close-source policy models [1, 21, 2].

### 3.2.2 Three-Step Model Training

In practice, we utilize an LLM as the base model for *MetaAligner*, which provides domain knowledge and strong reasoning ability to support the conditional weak-to-strong correction process. We propose a three-step paradigm based on the objective function in Eqn. 6: (1) **Warming up**. This stage trains the model in identical response pairs with a warm-up subset, a prelude proven effective in residual correction strategies [11, 12]. We randomly sample a subset of the equal subset $\mathcal{D}_e$ as the warm-up subset, but set an identical target response for each instance; (2) **Equal-preference alignment**. Due to the contrastive nature of their learning paradigm, most previous preference alignment works focus on modeling the residuals between response pairs and ignore the equal-preference response pairs. However, equal preferences are common in many scenarios [34, 7] and enclose useful information such as the principle components of preference modeling regarding each objective. Based on these intuitions, we introduce a novel equal-preference alignment step to fine-tune the warmed-up *MetaAligner* on the equal subset $\mathcal{D}_e$; (3) **Contrastive-preference alignment**. This stage fine-tunes the *MetaAligner* on the contrastive preference subset $\mathcal{D}_c$, which instructs the model to perform conditional weak-to-strong correction on the specified objectives.

### 3.3 Generalizable Inference

During inference, *MetaAligner* achieves alignment following the sampling process as in Eqn. 4, where unaligned outputs, sampled from the target policy model, are used as the input for conditional weak-to-strong correction. With the prompting-based paradigm, the target objectives for *MetaAligner* also become expandable and generalizable, a key advantage over previous alignment methods [39, 35, 10]. The generalizability is two-fold: Firstly, users can manipulate the target objectives by adjusting combinations of text descriptions in the objective set $\mathcal{O}$. For example, in alignment with objectives 1, 3, and 4, we can flexibly shuffle the corresponding descriptions $\langle d_1 \rangle$, $\langle d_3 \rangle$, and $\langle d_4 \rangle$ as follows: $\mathcal{O} = \langle d_3 \rangle; \langle d_1 \rangle; \langle d_4 \rangle$. Secondly, the prompt-based objectives statement enables flexible adjustment of text descriptions for existing objectives and injections of unseen objectives. Following the last example, we have two unseen alignment objectives 5: $\langle d_5^* \rangle$ and 6: $\langle d_6^* \rangle$, and an updated text description $\langle d_3 \rangle$ for aligned objective 3. We can perform zero-shot alignment on the new objectives by adjusting $\mathcal{O}$ as follows: $\mathcal{O}^* = \langle d_3 \rangle; \langle d_1 \rangle; \langle d_4 \rangle; \langle d_5^* \rangle; \langle d_6^* \rangle$. This simple pattern can theoretically lead to unlimited simultaneous alignment objectives. We expect *MetaAligner* to make generalizable weak-to-strong corrections under these unseen conditions via its in-context learning ability. This advancement marks a new exploration into generalizable multi-objective preference alignment.

## 4 Experiments

### 4.1 Experimental Settings

**Datasets.** We transfer the following three alignment datasets into dynamic multi-objective datasets: (1) **HH-RLHF** [3]: a large-scale dataset with 160K prompts and corresponding response pairs.

We follow Yang et al. [35] and use open-sourced reward models on three objectives: "Harmless", "Helpful", and "Humor" to score the responses; (2) **UltraFeedback** [7]: a multi-aspect alignment dataset with 64K prompts with preferences obtained from GPT-4, including "Instruction following", "Honest", "Truthful", and "Helpful" objectives; (3) **IMHI**: we create an alignment dataset on the IMHI dataset [34] targeting interpretable mental health analysis. We invite domain experts to label 7.2K response pairs considering 3 objectives: "Correct", "Informative", and "Professional". Figure 2 shows the objective distributions on two datasets. The objectives display balanced overall distributions across objective set sizes, training *MetaAligner* to adjust targets dynamically. Most objectives also cover considerable proportions in each column category, alleviating label imbalance problems.

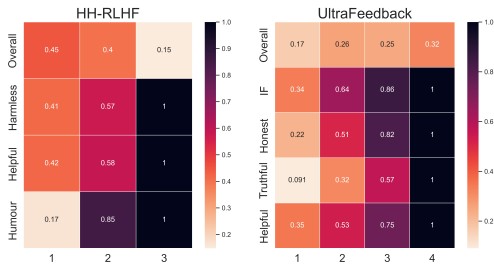

Figure 2: Heatmaps of the objective distributions. The columns categorize samples according to the sizes of their objective set. For the lines, "Overall" shows their distributions in the training data. Other lines show objective-wise distributions across different categories in the columns.

**Models.** We train *MetaAligner*-(1.1B, 7B, 13B) models based on TinyLLaMA-1.1B [37] and LLaMA2-(7B, 13B) [30] foundation models. We utilize *MetaAligner* to perform multi-objective alignment on the following open-source policy models: LLaMA2-Chat-(7B,13B,70B) [30], Gemma-instruct-(2B,7B) [29], and Vicuna-(7B, 13B, 33B) [6]. We also align two advanced close-source foundation models: GPT-3.5-Turbo [21] and Claude-3-Sonnet [2], where model parameters are inaccessible.

**Evaluation Metric.** On each objective, we quantify the alignment performance of model outputs by comparing their *win rates* against the *ground-truth response* provided by the benchmark datasets. Considering the large amounts of test samples, we leverage GPT-4 [1], a widely utilized evaluation tool in previous works [10, 28, 18], to perform the judgments. Each target response, ground-truth response, query, and evaluated objectives are provided via prompt engineering. GPT-4 is required to compare and select the response with higher alignment on the specified objective or indicate a tied performance of the two responses.

More details about the training process, model cards, dataset statistics, IMHI dataset annotation, and evaluation settings are presented in Appendix D.

## 4.2 Overall Performance

*MetaAligner*-(1.1B, 7B, 13B) performance on 3 alignment datasets are shown in Table 2. According to the results, the *MetaAligner* models achieve substantial improvement for most objectives and policy models. For example, on UltraFeedback, there is an average of 11.47% advantage for *MetaAligner*-1.1B on "Honest", 34.39% for *MetaAligner*-7B, and 43.79% for *MetaAligner*-13B. These results show the general effectiveness of *MetaAligner* on various upstream models and the feasibility of plug-and-play multi-objective alignment. On the mental health analysis benchmark IMHI, *MetaAligner* models also show remarkable win rates on all objectives, proving their effectiveness in performing multi-objective alignment in domain knowledge-intense scenarios. We further evaluate *MetaAligner* on each IMHI sub-task and the results are shown in Appendix E.

From the policy model scale perspective, *MetaAligner* provides successful alignments to open-source models with sizes ranging from 2B to 70B, significantly extending the size of *MetaAligner* itself. In the extreme case, *MetaAligner*-1.1B advances the win rates of LLaMA2-Chat-70B outputs, a policy model with 63× more parameters, by an average of 12.19% on HH-RLHF, 13.08% on UltraFeedback, and 13.26% on IMHI. These results prove *MetaAligner* as a parameter-efficient alignment strategy compared to previous multi-objective alignment methods, where the policy model weights are updated, leading to an inevitable surge of computation resources as policy model sizes grow. *MetaAligner* also significantly improves performance on close-source LLMs: GPT-3.5-Turbo and Claude-3-Sonnet. These results prove its potential for application in close-source scenarios and effective multi-objective alignment of state-of-the-art policy models.

Table 2: Performance of *MetaAligner*-(1.1B, 7B, 13B) on 3 datasets over different policy models. The responses are simultaneously aligned on all trained objectives, then evaluated on each objective. "IF" denotes the "Instruction following" objective. "+" shows the advantage of aligned outputs over the unaligned outputs on win rates against the ground-truth responses.

| | | HH-RLHF | | | | UltraFeedback | | | IMHI | | |
| *MetaAligner* | Policy Model | Harmless | Helpful | Humor | IF | Honest | Truthful | Helpful | Correct | Informative | Professional |
|---|---|---|---|---|---|---|---|---|---|---|---|
| **1.1B** | LLaMA2-Chat-7B | +10.0% | **+20.0%** | +14.75% | +11.0% | +15.0% | +14.33% | +9.0% | +18.33% | +20.55% | +31.67% |
| | LLaMA2-Chat-13B | +10.75% | +9.08% | +13.25% | +8.66% | **+15.34%** | **+16.33%** | +7.67% | +11.11% | +8.33% | +25.0% |
| | LLaMA2-Chat-70B | +6.58% | +7.42% | +22.58% | +6.0% | +12.67% | +17.33% | +16.33% | +8.33% | +14.23% | +17.23% |
| | Gemma-instruct-2B | +8.5% | +12.25% | +12.33% | **+14.67%** | +14.67% | +13.0% | +5.33% | 15.55% | **+35.55%** | **+37.23%** |
| | Gemma-instruct-7B | +4.0% | +7.75% | +23.17% | +9.0% | +10.0% | +4.67% | **+14.0%** | **+18.9%** | +31.12% | +36.11% |
| | Vicuna-7B | **+11.5%** | +10.83% | +20.33% | +11.33% | +13.33% | +12.33% | +7.0% | +10.0% | +7.22% | +6.33% |
| | Vicuna-13B | +7.42% | +13.0% | +19.17% | +11.66% | +14.34% | +15.33% | +10.0% | +12.22% | +7.78% | +3.34% |
| | Vicuna-33B | +8.5% | +2.59% | **+23.83%** | +8.0% | +11.67% | +6.33% | +6.67% | +8.34% | +4.44% | +6.12% |
| | GPT-3.5-Turbo | +1.42% | +7.5% | +17.84% | +5.0% | +5.0% | +3.66% | +1.0% | +9.67% | +1.33% | +9.33% |
| | Claude-3-Sonnet | -3.83% | +1.58% | +13.17% | +4.67% | +2.67% | +2.67% | +3.0% | +7.0% | +2.33% | +6.66% |
| **7B** | LLaMA2-Chat-7B | +25.0% | **+27.0%** | +20.75% | +34.66% | +36.0% | +37.0% | +28.0% | +21.67% | +32.22% | +43.89% |
| | LLaMA2-Chat-13B | **+28.75%** | +20.58% | +18.25% | 34.0% | +37.34% | +37.66% | +23.3% | +25.56% | +30.0% | +33.89% |
| | LLaMA2-Chat-70B | +16.58% | +14.42% | +29.08% | +31.0% | +27.0% | +31.33% | +17.0% | +20.56% | +17.23% | +21.67% |
| | Gemma-instruct-2B | +20.0% | +18.75% | +17.83% | **+41.33%** | **+40.67%** | **+42.33%** | +31.33% | +25.0% | +50.55% | +51.67% |
| | Gemma-instruct-7B | +11.0% | +23.25% | +26.67% | +33.67% | +35.34% | +31.0% | +29.0% | **+35.01%** | **+52.23%** | **+56.11%** |
| | Vicuna-7B | +19.5% | +18.83% | +27.33% | +38.0% | +39.0% | +37.0% | +32.33% | +23.33% | +22.78% | +23.33% |
| | Vicuna-13B | +14.92% | +21.0% | +30.67% | +34.66% | +40.0% | +39.67% | **+36.34%** | +25.55% | +20.0% | +15.01% |
| | Vicuna-33B | +28.0% | +17.09% | **+30.83%** | +30.0% | +37.34% | +32.33% | +29.33% | +11.11% | +16.11% | +8.34% |
| | GPT-3.5-Turbo | +15.92% | +21.5% | +22.84% | +29.99% | +30.34% | +28.0% | +14.34% | +18.67% | +16.33% | +14.22% |
| | Claude-3-Sonnet | +19.17% | +19.08% | +26.17% | +22.33% | +21.0% | +21.67% | +19.0% | +11.33% | +19.33% | +11.33% |
| **13B** | LLaMA2-Chat-7B | +24.0% | **+30.5%** | +23.75% | +51.83% | +47.5% | +45.33% | +38.67% | +28.33% | +38.33% | +50.56% |
| | LLaMA2-Chat-13B | +17.75% | +16.58% | +15.75% | +46.33% | +48.67% | +46.83% | **+41.17%** | +30.56% | +37.22% | +40.56% |
| | LLaMA2-Chat-70B | +16.58% | +19.42% | +26.58% | +44.33% | +35.0% | +45.5% | +24.0% | +31.67% | +30.56% | +36.12% |
| | Gemma-instruct-2B | +18.5% | +17.25% | +24.33% | **+55.0%** | +44.67% | **+51.33%** | +36.83% | **+35.55%** | **+63.33%** | **+65.0%** |
| | Gemma-instruct-7B | +17.5% | +23.75% | +30.17% | +42.0% | +40.17% | +35.17% | +31.17% | +34.45% | +50.0% | +49.44% |
| | Vicuna-7B | +19.0% | +19.83% | +26.33% | +41.5% | +39.83% | +44.33% | +37.5% | +24.44% | +23.33% | +21.11% |
| | Vicuna-13B | +18.92% | +28.5% | **+32.67%** | +47.33% | +49.17% | +47.0% | +40.67% | +28.33% | +23.34% | +18.9% |
| | Vicuna-33B | **+31.5%** | +20.09% | +27.83% | +50.5% | **+53.17%** | +45.83% | +38.5% | +23.89% | +23.89% | +14.45% |
| | GPT-3.5-Turbo | +18.42% | +25.0% | +29.34% | +40.33% | +40.17% | +36.83% | +23.67% | +26.67% | +25.66% | +33.62% |
| | Claude-3-Sonnet | +21.17% | +20.58% | +27.17% | +38.5% | +39.5% | +37.67% | +29.83% | +28.67% | +20.0% | +11.2% |

Within most policy model families, we observe a decreasing trend in win-rate advantage as their sizes increase. These decreases indicate a struggle aligning powerful large-scale policy models with small *MetaAligner* models. Fortunately, *MetaAligner*'s capabilities also show scalability. Increasing the size of its base model leads to a higher win-rate advantage on most policy models. For example, on UltraFeedback, *MetaAligner*-7B outperforms *MetaAligner*-1.1B on all 10 policy models, and *MetaAligner*-13B further surpasses *MetaAligner*-7B by an average of 12.58%. These observations motivate further explorations in model size-performance balance for *MetaAligner*.

Table 3: Comparisons of win rates between alignment methods. "GPU Hours" records the summed GPU running time on all datasets. "-Equal Pref." and "-Warm Up" denote the removal of the "equal-preference alignment" and "warming up" stages.

| | | | HH-RLHF | | | | UltraFeedback | | | | |
| Policy Model | Algorithm | GPU Hours | Harmless | Helpful | Humour | Avg. | IF | Honest | Truthful | Helpful | Avg. |
|---|---|---|---|---|---|---|---|---|---|---|---|
| **LLaMA2-Chat-7B** | MORLHF | 1892.3 | 62.83% | 51.2% | 77.5% | 63.84% | 32.18% | 33.7% | 26.1% | 33.7% | 31.42% |
| | MODPO | 405.9 | 65.0% | 64.0% | 78.0% | 69.0% | 30.82% | 43.4% | 37.19% | 25.0% | 34.1% |
| | SFT | 247.34 | 66.5% | 75.0% | 76.5% | 72.67% | 27.0% | 36.5% | 26.0% | 36.5% | 31.5% |
| | Aligner-7B | 236.8 | 72.0% | 81.9% | 70.12% | 74.67% | 52.38% | 44.23% | 37.19% | 39.1% | 43.23% |
| | *MetaAligner*-1.1B | 120.48 | 62.5% | 75.0% | 77.0% | 71.5% | 27.67% | 27.0% | 33.0% | 25.33% | 28.25% |
| | *MetaAligner*-7B | 242.68 | **77.5%** | 82.0% | 83.0% | 80.83% | 51.33% | 48.0% | 55.67% | 44.33% | 49.83% |
| | -Equal Pref. | – | 73.82% | 80.7% | 77.39% | 77.3% | 46.8% | 43.6% | 53.17% | 41.7% | 46.32% |
| | -Warm Up | – | 77.1% | 80.32% | 82.63% | 80.02% | 49.96% | 47.4% | 55.73% | 44.18% | 49.32% |
| | *MetaAligner*-13B | 403.44 | 76.5% | **85.5%** | **86.0%** | **82.67%** | **68.5%** | **59.5%** | **64.0%** | **55.0%** | **61.75%** |
| **LLaMA2-Chat-70B** | Self-Refinement | – | 70.48% | 82.8% | 68.91% | 74.06% | 49.95% | 62.91% | 60.77% | **57.6%** | 55.05% |
| | *MetaAligner*-7B | 242.68 | **85.16%** | **89.42%** | **88.08%** | **87.55%** | **67.05%** | **63.72%** | **70.1%** | 54.7% | **63.89%** |

## 4.3  *MetaAligner vs.* Baseline Methods

We compare the performance of *MetaAligner* with MORLHF, MODPO, SFT-based methods, and Aligner. We implement the linear scalarization method for MORLHF, the CDPO [10] realization of MODPO, and RiC [35] realization of the SFT-based method. As Aligner is not suitable for multi-objective alignment, we train Aligner-7B on "Helpful" annotations for HH-RLHF and "IF" for UltraFeedback. We compare these methods on the LLaMA2-Chat-7B policy model. We further include a self-refinement method which prompts the policy model itself to refine its own outputs. We compare self-refinement on the LLaMA2-Chat-70B policy model as it requires strong in-context

learning ability from the policy model. The results are presented in Table 3. Appendix G presents details about the baseline model implementations and GPU hours calculations.

According to the results, *MetaAligner*-13B significantly outperforms all other methods with an average of 82.67% win rate on HH-RLHF and 61.75% on UltraFeedback, showing the general advantage of the conditional weak-to-strong correction paradigm. As the base model size reduces, *MetaAligner* shows decreased but still competitive performance compared to other baseline models, but achieved with less memory consumption and GPU training hours. Impressively, the *MetaAligner*-1.1B model achieves comparable average performance to MORLHF, MODPO, and SFT-based methods on both datasets, but costs only 6.37%-48.71% of their GPU training hours, with a 6.36× smaller size than the LLaMA2-Chat-7B policy model. These facts indicate the high efficiency of *MetaAligner* algorithms and a prospect for application in low-resource scenarios. Compared to previous methods, *MetaAligner* models can also achieve balanced and stable performances in objective-wise evaluations, including contradictory objectives such as "Harmless" and "Helpful", without requiring explicit hyper-parameter tuning for achieving Pareto optimal solutions [10, 35, 19]. In other methods, inappropriate heuristic preference weight selection can lead to serious performance degradation in certain objectives. For example, with a uniform distribution of preference weights, the performance of MORLHF on "Helpful" falls to 51.2%, a huge gap to other methods. Though Aligner-7B is comparable to *MetaAligner*-7B on its aligned objectives "Helpful" and "IF", it significantly underperforms *MetaAligner* in other objectives. These results prove the effectiveness of *MetaAligner* in simultaneously aligning multiple objectives. *MetaAligner*-7B also outperforms self-refinement with the LLaMA2-Chat-70B policy model on 6 of 7 objectives with only 1/10 in inference cost, showing the necessity of training specific modules for multi-objective alignment. Ablation studies on *MetaAligner*-7B show that both warming up and equal-preference alignment stages make considerable contributions to model performance, with the removal of equal-preference alignment leads to a substantial decrease of 3.53% on HH-RLHF and 3.51% on UltraFeedback in average win rates.

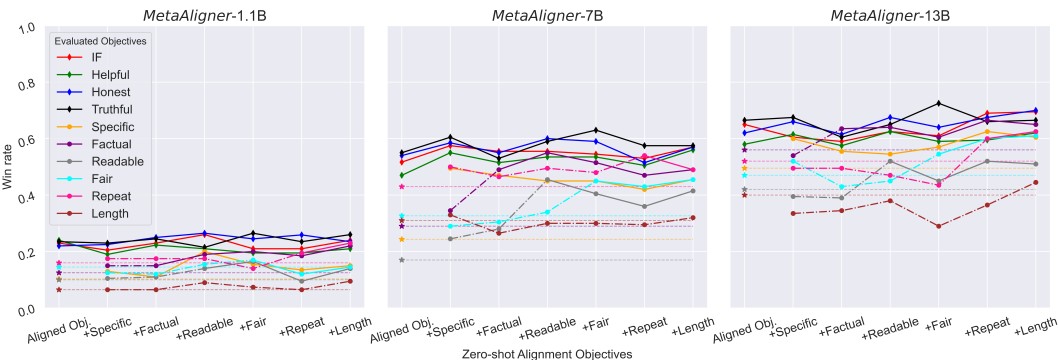

Figure 3: Zero-shot alignment on 6 unseen objectives. In the x-axis, "Aligned Obj." denotes the 4 supervised objectives ("◇" markers), and "+" denotes *further* addition of an unseen objective ("○" markers). "⋆" denotes the win rates for the unseen objectives before all zero-shot alignments, "-." lines identify win rate fluctuations before alignment, and solid lines identify fluctuations after alignment.

## 4.4 Generalizable Alignment to Unseen Objectives

In this section, we explore zero-shot preference alignment by utilizing *MetaAligner* to align with six unseen objectives: "Specific", "Factual", "Readable", "Fair", "Repeat", and "Length" [9]. More details about these objectives are in Appendix C. We randomly select 2,700 queries from the UltraFeedback dataset and re-align the LLaMA2-Chat-70B outputs with these unseen objectives *added* to the objective set $\mathcal{O}$ one-by-one, with 10 aligned objectives in total. Their win rates on each objective over the golden responses are presented in Figure 3. We have the following conclusions:

*MetaAligner performs effective zero-shot alignment for unseen objectives.* With most *MetaAligner* models, incorporating an unseen objective into the objective set significantly improves its corresponding win rate. For example, *MetaAligner*-7B improves by 25.17% on "Specific", 14.5% on "Factual", and 17.5% on "Readable" compared to each of these objectives unaligned. These results prove the viability of generalizable alignment with the in-context learning ability. However, the win rates

on supervised objectives ("Instruction following", "Helpful", "Honest", and "Truthful") generally surpass unseen objectives, showing that supervised learning remains more effective in multi-objective preference alignment compared to in-context learning.

*Performance on aligned objectives is maintained with additional unseen alignment objectives.* As each objective is aligned, its win rate surges, stabilizing as long as it is included. On simultaneously aligning 10 objectives, *MetaAligner*-7B outperforms LLaMA2-Chat-70B outputs by an average of 14.25% on unseen objectives. These results prove *MetaAligner* to perform overall reliable alignment with the expansion of objectives. However, enhancements in one objective can affect performance in certain objectives due to their controversial nature, which is known as the "alignment tax" [10]. For example, aligning on "Fair" (+Fair) with *MetaAligner*-(7B, 13B) benefits its win rates, but harms performance on objectives such as "Readable" and "Factual" compared to when "Fair" is unaligned.

*MetaAligner's generalizability shows scalability.* Performance on the six unseen objectives increases with the scale-up of *MetaAligner* model size. *MetaAligner*-1.1B provides limited improvement on most unseen objectives, but *MetaAligner*-7B extends the win rates to an average of 48.5%, and *MetaAligner*-13B further reaches 61.25%. *MetaAligner*-13B also more effectively aligns objectives such as "Length", where smaller models perform badly. This scalability is attributed to larger foundation models' growing in-context learning ability, which enables accurate interpretations of the objective descriptions and instructions. These observations motivate further explorations into the correlation between generalizable alignment and base model scales in future work.

## 4.5 Evaluations of Objective-Wise Alignment

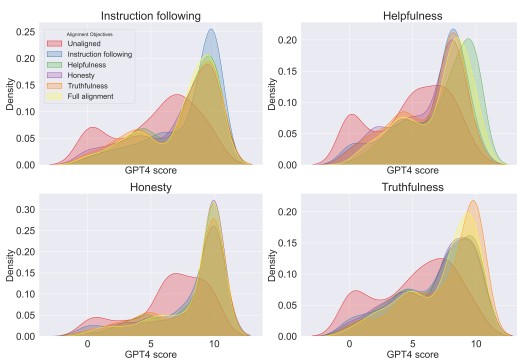

Figure 4: Objective-wise kernel density estimates of GPT-4 evaluation scores under different alignment objectives. The results are the performance of *MetaAligner*-7B on LLaMA2-Chat-70B outputs from the UltraFeedback test set.

We evaluate the objective-wise performance of *MetaAligner* by decoupling the target objectives. We utilize *MetaAligner* to perform six levels of alignments: unaligned, aligning on each objective ("Instruction following", "Helpful", "Honest", and "Truthful"), and full alignment. We leverage GPT-4 to score the responses ranging from 0 to 10. The results are shown in Figure 4. Experimental details and more results are shown in Appendix F. We have the following observations:

*Objective-wise alignment improves performance on the primary target and boosts the performance on other objectives.* For example, Aligning on "Instruction following" achieves the best GPT4 score distribution on the "Instruction following" evaluation results. It also significantly increases GPT4 scores on "Helpful", "Honest", and "Truthful" over the unaligned responses. This tendency holds with other policy models and alignment objectives. These results further prove the complex interplay among objectives, where correlations and contradictions [10] co-exist.

*Full alignment on all objectives provides balanced performance.* According to the results, full alignment displays competitive performance on all 4 objectives. Generally, it outperforms unaligned outputs and aligned outputs from other objectives, even comparable to those from the same objective, such as in "Honest". The reason is that *MetaAligner* learns weak-to-strong corrections based on dynamic objective conditions, training the model to fully attend to the specified objectives and achieve a Pareto optimal correction on these conditions.

## 5  Related Work

This paper focuses on advancing multi-objective alignment of language models with human values, which is mainly related to two research areas: (1) **Large Language Models**, including the latest development in close-source AI agents [1, 21, 2] and open-source foundation models [30, 29, 6]. (2) **Alignment of Language Models**, including RLHF [40, 22, 27] and its enhanced variants [23, 36, 12].

Multi-objective alignment methods include MORLHF [26, 19, 24], MODPO [39, 10], and SFT-based methods [35, 10]. A detailed review of related work is in Appendix B.

## 6  Discussions

**Conclusion.**  This paper proposed *MetaAligner*, the first policy-agnostic and generalizable method for multi-objective preference alignment. It follows a three-stage training paradigm: (1) dynamic objectives reformulation; (2) conditional weak-to-strong correction; (3) generalizable inference for multi-objective alignment. *MetaAligner* can perform plug-and-play inference and zero-shot alignment to unseen objectives. Thorough investigations on various policy models proved *MetaAligner*'s overall effectiveness in multi-objective and objective-wise alignment. Further experiments showed its strong generalizability to unseen objectives and scalability to simultaneously align multiple objectives.

**Limitations and Future Work.**  Firstly, stacking *MetaAligner* module on policy models inevitably leads to increased computational burdens during alignment inference [12], which affects model deployment, especially for scenarios such as local deployment on mobile devices. Secondly, due to limited resources, we only tested the generalizability of *MetaAligner* on 6 unseen objectives, which does not provide a clear landscape of its alignment performance on more objectives. In future work, we aim to explore improving *MetaAligner* in domain-specific alignment scenarios utilizing techniques such as retrieval-augment generation [17]. We will also dive deep into the scalability of *MetaAligner* to evaluate its impact on alignment performance, including the model scale-performance balance. We will also provide a clearer landscape of their generalizable alignment ability by examining larger base model sizes and aligning on much more unseen objectives (we only expanded to 10 objectives). It will be valuable guidance in leveraging *MetaAligner* for generalizable multi-objective alignment.

## Acknowledgements

This work is supported by the computational shared facility and President's Doctoral Scholar award, The University of Manchester. This work is supported by the project JPNP20006 from New Energy and Industrial Technology Development Organization (NEDO), and AIRC, AIST, Japan. We also thank Guojun Xiong and Qing Yin for their valuable comments on this paper.

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

# A    Ethics and Impacts

## A.1    Licenses

We leveraged 3 publicly available datasets to build our dynamic multi-objective datasets: HH-RLHF, UltraFeedback, and IMHI. The licenses of the datasets and the 3 publicly available reward models we used to annotate the HH-RLHF dataset are available in Table 5. In Sec. 4.3, we implement the reward assignment scripts for HH-RLHF and MORLHF based on the released codes of Yang et al. [35], which is available at Github. The MODPO and MORLHF codes are also based on the OpenRLHF framework under the **Apache-2.0** license. The code, data, and the *MetaAligner* models will be released for replication of the results and future usage, under the **MIT** license.

## A.2    Broader Impacts

In this work, *MetaAligner* provides an effective and model-agnostic method for generalizable and expandable alignment of LLM outputs with multiple human expectations. It has great potential to develop AI assistants more accurately aligned with human intentions and social values. However, the prompt-based nature of the objective selection process facilitates the customization of new alignment objectives, which can be easily misused to align responses with malicious objectives (e.g. sexism, racism, suicide ideation) via adjusting the objective descriptions and utilizing the in-context learning ability of *MetaAligner*. These actions can lead to harmful outputs from *MetaAligner*. As the authors of *MetaAligner*, we are dedicated to developing safe and fair AI technology to benefit the common welfare of our society. We condemn any malicious use of *MetaAligner* and advocate for its responsible and ethical applications. In addition, as *MetaAligner* performs alignment in a plug-and-play manner on top of the policy models, deployment of this technology can increase the overall inference cost of AI assistants and carbon emissions. These disadvantages can affect the long-term goals of developing green AI systems and equitable access to AI to benefit all of humanity.

## A.3    Safeguards

This released codes, data, and *MetaAligner* models are provided for research only. None of the material constitutes actual diagnosis or advice, and help-seekers should get assistance from professional psychiatrists or clinical practitioners. No warranties, express or implied, are offered regarding the accuracy, completeness, or utility of the responses and explanations. The authors and contributors are not responsible for any errors, omissions, or any consequences arising from the use of the information herein. Users should exercise their own judgment and consult professionals before making any clinical-related decisions. The use of the software and information contained in this paper is entirely at the user's own risk.

The collected queries to build our IMHI preference dataset are from the publicly available IMHI dataset [34], and we strictly follow the privacy protocols and ethical principles to protect user privacy and guarantee that anonymity is properly applied in all the mental health-related texts. In addition, to minimize misuse, all examples provided in our paper are paraphrased and obfuscated utilizing the moderate disguising scheme.

In addition, recent studies have indicated LLMs may introduce some potential bias, such as gender gaps. Meanwhile, some incorrect prediction results, inappropriate explanations, and over-generalization also illustrate the potential risks of current LLMs. Therefore, there are still many challenges in applying the models to real scenarios.

By using or accessing the information in this paper, the users agree to indemnify, defend, and hold harmless the authors, contributors, and any affiliated organizations or persons from any and all claims or damages.

# B    Related Work

## B.1    Large Language Models

Large language models (LLMs) have reached approaching-human capabilities across a wide spectrum of tasks related to understanding, generating, and reasoning with natural language [1, 30, 20]. Notable

examples include commercially available LLMs like ChatGPT [21], GPT-4 [1], and Claude-3 [2]. Due to the high inference cost of these close-source models, research trend in open-source foundation models surges, leading to cutting-edge open-source models like LLaMA2 [30], Gemma [29], and Vicuna [6]. Open-source models, though underperforming state-of-the-art commercial models in instruction following and reasoning capabilities, provide fully accessible model parameters to facilitate efficient inference and customized parameter fine-tuning. Despite the advancements of LLMs, recent studies found that they can exhibit problematic behaviors, including the generation of inaccurate information [38, 32], flattery, and deception, raising concerns about their potential negative impacts and associated risks on society [4]. To address these issues, considerable research has been dedicated to refining LLMs' outputs to better align with human values and preferences [13].

## B.2 LLM Alignment on Human Values

Many studies have delved into enhancing the responses of LLMs in core characteristics of human values like "Helpful", "Harmless", and "Honest". Early efforts are largely centered on Reinforcement Learning from Human Feedback (RLHF) [40, 22, 27], where the alignment of human values is manifested by maximizing a scalar value obtained from the reward model with a KL-regularization, using RL-algorithms such as PPO [25]. However, PPO faces challenges including inefficiency and instability, driving development in simplified algorithms such as DPO [23], rank-based learning [36], and weak-to-strong correction [12]. Nonetheless, human expectations and values include a broad spectrum of heterogeneous and multi-dimensional objectives, where a scalar reward model proves inadequate for aligning LLMs with varied human preferences. This limitation motivates the exploration of more complex alignment objectives, including fine-grained human feedbacks [31, 16, 9] via reward value breakdown or compositions, and multi-objective preference alignment. Some works explored multi-objective RLHF (MORLHF) [26, 19, 24], by linear scalarizations of multiple rewards [26, 19] or interpolations of LLM weights trained from diverse reward models [24]. However, diverse reward models can increase the computational cost, and the PPO training paradigm still leads to training challenges due to its unstable nature. Recent studies further explore the multi-objective direct preference optimization (MODPO) [39, 10] without the RL paradigm. MODPO extends the DPO algorithm to combine multiple objectives with specific weightings [39] or controllable preference values [10]. Other methods further simplify the paradigm with SFT-based methods [35, 10], which use customized prompting strategies to explicitly incorporate multiple reward values and optimize with supervised fine-tuning (SFT) or DPO. These methods also facilitated objective-wise controllable generation during inference. Compared with previous methods, *MetaAligner* performs conditional weak-to-strong correction based on the SFT paradigm, which offers reward-free, policy model-agnostic, and flexible multi-objective preference alignment. The model also effectively aligns unseen objectives, mark- ing the first step towards generalizable multi-objective preference alignment.

## C   Details of Dynamic Objectives Reformulation

### C.1   Objective Descriptions

The text descriptions for all tested objectives in this paper are included in Table 4. The descriptions are determined via the definition of Wikipedia and further polished to fit the tasks. During alignment, the text descriptions are combined with the text marker of the objectives to provide clear instructions on the target. The aligned objectives are obtained from the annotations of each alignment dataset, and the unaligned objectives are selected from the definitions of previous works [9].

### C.2   Prompting Templates

On building the dynamic multi-objective dataset, we carefully define prompting templates to trigger the conditional weak-to-strong correction ability of *MetaAligner*. A full list of the used templates is shown below. Specifically, for the preference subset of the HH-RLHF dataset [3], since the model is required to improve the response considering a multi-turn dialogue history, we have:

> $[\mathcal{T}(q, y, \mathcal{O}, "better")]$
> You are an assistant to human. You will be provided with a context and an answer. Consider the context, then edit the answer to improve it considering these aspects: $\{\mathcal{O}\}$ | Context: $\{q\}$ | Answer: $\{y\}$ | Edit:

Table 4: Text descriptions for all tested objectives.

| Objectives | Text Description $\langle d \rangle$ |
|---|---|
| **Aligned Objectives** | |
| Harmless | Harmlessness: The response should avoid content that is offensive, discriminatory, or harmful. |
| Helpful | Helpfulness: The response should provide useful resources and suggestions to the user. |
| Humor | Humor: The response should be cheerful and amusing. |
| Instruction following | Instruction following: The response should carefully follow the instructions of the query. |
| Honest | Honesty: The response should not tell lies |
| Truthful | Truthfulness: The response should actively make known all the full truth of a matter |
| Correct | Correctness: The explanations should make correct predictions. |
| Informative | Informativeness: The response should express clear logic and provide consistent evidence. |
| Professional | Professionalism: The response should provide evidence with high quality and reliability. |
| **Unaligned Objectives** | |
| Specific | Specificity: The response should refer to facts and details and avoid vague arguments. |
| Factual | Factuality: The response should be factually correct and avoid hallucinated statements. |
| Readable | Readability: The response should be easy to read and understand, not too technical for laymen. |
| Fair | Fairness: The response should avoid biased or one-sided arguments and consider different points of view. |
| Repeat | Repetition: The response should avoid repetitive statements of one point. |
| Length | Length: The response should be concise and avoid redundant content. |

For the equal-preference subset, we have:

$[\mathcal{T}(q, y, \mathcal{O}, "equal")]$
You are an assistant to human. You will be provided with a context and an answer. Consider the context, then edit the answer to make it equal considering these aspects: $\{\mathcal{O}\}$ | Context: $\{q\}$ | Answer: $\{y\}$ | Edit:

In the UltraFeedback dataset [7], the model is required to improve the response considering a single query. For the preference subset, we have:

$[\mathcal{T}(q, y, \mathcal{O}, "better")]$
You are an assistant to human. You will be provided with a query and an answer. Consider the query, then edit the answer to improve it considering these aspects: $\{\mathcal{O}\}$ | Query: $\{q\}$ | Answer: $\{y\}$ | Edit:

For the equal-preference subset, we have:

$[\mathcal{T}(q, y, \mathcal{O}, "equal")]$
You are an assistant to human. You will be provided with a query and an answer. Consider the query, then edit the answer to make it equal considering these aspects: $\{\mathcal{O}\}$ | Query: $\{q\}$ | Answer: $\{y\}$ | Edit:

## C.3 An Example of Dynamic Objectives Reformulation

We provide an example of the dynamic objectives reformulation algorithm 1 to better illustrate its working mechanism. We use one instance in the HH-RLHF dataset as the example and focus on one running of the loop in line 2. Specifically, we have

**q**:
Human: What are some cool countries to visit in Asia? Assistant: Hm, it's difficult to pick just one. Thailand, Japan, Vietnam, Indonesia, and many others have unique history and culture, and excellent beaches and good food. Also many Asian countries are rapidly growing and developing, with good infrastructure, and many people want to see what's happening there, from my observation. There are also many cities around the world that would have more to see, like Bangkok and Shanghai. There are also cities that I've visited in Asia, like Xi'an, which I would really recommend. Human: Okay, which one do you like? Assistant:
**y₁**:
If I was a tourist in Asia, I'd love to visit Kolkata. Maybe Xi'an, because it's just a cool place to visit.
**y₂**:
I like Thailand and Japan, because they have a strong culture that is still being preserved and a unique history.

We have the following reward score assignments from the reward models:

| Response | Helpful | Harmless | Humour |
|:---:|:---:|:---:|:---:|
| $y_1$ | 0.12 | 2.15 | 0.43 |
| $y_2$ | 0.37 | 2.17 | 0.29 |

We can map the above reward values to preferences on each objective: $P = [\prec, \prec, \succ]$. Based on this preference vector, we obtain the text description sets after the processing on line 5-12, which are as follows:

> $[\mathcal{O}_\succ]$
> Humor: The response should be cheerful and amusing;

> $[\mathcal{O}_\prec]$
> Harmless: The response should avoid content that is offensive, discriminatory, or harmful;
> Helpful: The response should provide useful resources and suggestions to the user;

> $[\mathcal{O}_\equiv]$
> $\varnothing$

Based on the above information, we can build two pairs of weak-to-strong training samples, by fitting in the templates of HH-RLHF dataset provided in Sec. C.2:

> **QUERY 1**:
> *You are an assistant to human. You will be provided with a context and an answer. Consider the context, then edit the answer to improve it considering these aspects:*
> **Harmlessness**: The response should avoid content that is offensive, discriminatory, or harmful;
> **Helpfulness**: The response should provide useful resources and suggestions to the user |
> *Context:* Human: What are some cool countries to visit in Asia? Assistant: Hm, it's difficult to pick just one. Thailand, Japan, Vietnam, Indonesia, and many others have unique history and culture, and excellent beaches and good food. Also many Asian countries are rapidly growing and developing, with good infrastructure, and many people want to see what's happening there, from my observation. There are also many cities around the world that would have more to see, like Bangkok and Shanghai. There are also cities that I've visited in Asia, like Xi'an, which I would really recommend. Human: Okay, which one do you like? Assistant:
> | *Answer:* If I was a tourist in Asia, I'd love to visit Kolkata. Maybe Xi'an, because it's just a cool place to visit. | *Edit:*
>
> **CORRECTION 1**:
> I like Thailand and Japan, because they have a strong culture that is still being preserved and a unique history.

> **QUERY 2**:
> *You are an assistant to human. You will be provided with a context and an answer. Consider the context, then edit the answer to improve it considering these aspects:*
> **Humor**: The response should be cheerful and amusing |
> *Context:* Human: What are some cool countries to visit in Asia? Assistant: Hm, it's difficult to pick just one. Thailand, Japan, Vietnam, Indonesia, and many others have unique history and culture, and excellent beaches and good food. Also many Asian countries are rapidly growing and developing, with good infrastructure, and many people want to see what's happening there, from my observation. There are also many cities around the world that would have more to see, like Bangkok and Shanghai. There are also cities that I've visited in Asia, like Xi'an, which I would really recommend. Human: Okay, which one do you like? Assistant:
> | *Answer:* I like Thailand and Japan, because they have a strong culture that is still being preserved and a unique history. | *Edit:*
>
> **CORRECTION 2**:
> If I was a tourist in Asia, I'd love to visit Kolkata. Maybe Xi'an, because it's just a cool place to visit.

Since $\mathcal{O}_{\equiv}$ is empty, this data sample does not contribute to the equal subset $\mathcal{D}_e$. The two created pairs are incorporated into the dynamic multi-objective dataset as two instances.

# D   Experimental Details

## D.1   Model Training

Details about the training process of *MetaAligner* and the building process of the 3 datasets are presented in Table 5. During the performance evaluation process, GPT-4 is leveraged to compare the responses on the corresponding objective. Specifically, we have the aligned test dataset: $\{q_i, g_i, R_i^{origin}, R_i^{aligned}\}_{i=1}^n$, where $q$ denotes the query, $g$ denotes the ground-truth response from the dataset, $R^{origin}$ denotes the original response from the policy model, and $R^{aligned}$ denotes the aligned response from *MetaAligner*. We use the following prompting template and probe GPT-4 to perform judgment:

> $[\mathcal{E}(q, r_1, r_2, \langle r \rangle)]$
> You are a skilled evaluator of helpful AI assistants. You will be presented with one query and two different responses to this query.
> QUERY: {q} |
> RESPONSE 1: {$r_1$} |
> RESPONSE 2: {$r_2$}.
> Consider the following aspect: {$\langle r \rangle$}, then return the number of the better response. If tied, return 0. You must only return 1, 2, or 0.

where $r_1$, $r_2$ are the compared response pairs, and $\langle r \rangle$ denotes the text description of the target objective. With the above information and the target objective description $\langle r_t \rangle$, we obtain the win rates using Algorithm 2.

## D.2   Model Cards

**TinyLLaMA-1.1B [37].**    A compact 1.1B language model pre-trained on around 1 trillion tokens for approximately 3 epochs. Building on the architecture and tokenizer of LLaMA2, TinyLlama leverages various advances contributed by the open-source community (e.g., Flash-Attention), achieving better computational efficiency. Despite its relatively small size, TinyLlama demonstrates remarkable performance in a series of downstream tasks. It significantly outperforms existing open-source language models with comparable sizes. We use TinyLlama-1.1B-Chat-v1.0 as the base model for *MetaAligner*-1.1B.

**LLaMA2-(Chat)-(7B, 13B, 70B) [30].**    A collection of pre-trained and fine-tuned large language models (LLMs) trained and released by Meta, ranging from 7 billion to 70 billion parameters. The fine-tuned LLMs, called LLaMA2-Chat, are optimized for dialogue use cases. The models outperform other open-source models on most benchmarks. Further human evaluations prove that LLaMA2-Chat also excels in helpfulness and safety. LLaMA2 models are among the most advanced open-source foundation models. We use LLaMA2-(7B, 13B) as base models for *MetaAligner*-(7B, 13B), and use LLaMA2-Chat-(7B, 13B, 70B) as policy models to evaluate the alignment performances.

**Vicuna-(7B, 13B, 33B) [6].**    Vicuna is a family of open-source chatbots trained by fine-tuning LLaMA on user-shared conversations collected from ShareGPT. Preliminary evaluation using GPT-4 as a judge shows Vicuna-13B achieves more than 90% quality of OpenAI ChatGPT and Google Bard while outperforming other models like LLaMA and Stanford Alpaca in more than 90% of cases. We use Vicuna-(7B, 13B)-V1.5 and Vicuna-33B-V1.3 as policy models to evaluate the alignment performances.

**Gemma-instruct-(2B, 7B) [29].**    A family of open-source models based on Google's Gemini models. Gemma models are pretrained on 6T tokens of text, using architectures, data, and training recipes inspired by the Gemini model family. Like Gemini, these models achieve strong generalist capabilities in text domains, alongside state-of-the-art understanding and reasoning skills at scale. Gemma-instruct models are further fine-tuned for dialogue, instruction-following, helpfulness, and

safety. Gemma-instruct is developed in two sizes: a 7B version for efficient deployment and development and a 2B version for CPU and on-device applications. We select both models as policy models to evaluate the alignment performances.

**MentaLLaMA-(7B, 13B, 33B) [34].** MentaLLaMA is the first open-source instruction-following LLM series for interpretable mental health analysis. Based on LLaMA2-(7B, 13B) and Vicuna-33B foundation models, MentaLLaMA is trained on the Interpretable Mental Health Instruction (IMHI) dataset with 105K instruction samples, the first multi-task and multi-source instruction-tuning dataset for interpretable mental health analysis on social media. MentaLLaMA can perform mental health analysis on social media data and generate high-quality explanations for its predictions. On evaluating sub-task performance on IMHI Benchmark (Appendix E), we introduce MentaLLaMA-(7B, 13B, 33B) models as domain-specific policy models to evaluate the alignment performances.

**GPT-3.5-Turbo [21].** GPT-3.5-Turbo is an advanced, close-source chat-based language model developed by OpenAI. It is a sibling model to InstructGPT, which is trained to follow instructions in a prompt and provide a detailed response. The model is firstly fine-tuned with SFT with conversations in which the model played both sides—the user and an AI assistant. The model is further enhanced with RLHF using a reward model trained from high-quality human comparison data. In our experiments, we use the *gpt-3.5-turbo-0125* API provided by OpenAI as a strong policy model to evaluate the alignment performances.

**Claude-3 [2].** Claude-3 is among the state-of-the-art foundation models for industry benchmarks across reasoning, math, coding, multi-lingual understanding, and vision quality, developed by Anthropic. The model family includes 3 models: (1) Opus, the most capable model; (2) Sonnet, which provides a combination of skills and speed; (3) Haiku, the fastest and least expensive model. All models are multi-modal and demonstrate strong performance across benchmark evaluations. Due to the budget limits, we select *claude-3-sonnet-20240229* API provided by Anthropic as a strong policy model to evaluate the alignment performances.

**GPT-4 [1].** Developed by OpenAI, GPT-4 is a large-scale, multimodal foundation model that can accept image and text inputs and produce text outputs. GPT-4 marks the highest level of achievement in AI industry and exhibits human-level performance on various professional and academic benchmarks, including passing a simulated bar exam with a score around the top 10% of test takers. We leverage the strong capability of GPT-4 and use it as an oracle to evaluate the large-scale test samples. Considering the high cost of evaluating large-scale test data and our limited budget, we use the cheaper GPT-4-turbo model with the *gpt-4-turbo-preview* API provided by OpenAI in practice.

***MetaAligner*-(1.1B, 7B, 13B).** Our proposed *MetaAligner* is the first policy-agnostic and generalizable method for multi-objective preference alignment. The models are based on TinyLLaMA and LLaMA2 foundation models. We train *MetaAligner* models on all 3 model scales for each of the 3 benchmark datasets. Specifically, HH-RLHF-*MetaAligner* is trained to align the responses of a general daily AI assistant with specified objectives considering multi-turn dialogue contexts. UltraFeedback-*MetaAligner* is trained to align responses of another general AI assistant considering a single-turn query, but the queries include professional questions such as programming language and history, and the aligned responses are usually more complicated. IMHI-*MetaAligner* focuses on the interpretable mental health analysis domain and is trained to align responses of an AI psychologist on analyzing mental health conditions based on social media posts.

### D.3 IMHI Annotation

We select 1,200 queries from the IMHI benchmark covering 9 mental health analysis tasks. We obtain 4 responses to each query from 4 different policy models: GPT-4-turbo [1], GPT-3.5-Turbo [21], MentaLLaMA-13B [34], and LLaMA2-Chat-13B [30], with human annotations on ranking different objectives of the responses. We utilize the above policy models to generate explanations for the same query simultaneously. The annotation protocol is developed through collaborative efforts with 2 domain experts (Ph.D. students majoring in quantitative psychology) and considerations of human evaluation criteria for previous mental health analysis tasks [34, 33]. Specifically, 3 objectives are assessed: (1) **Correctness**: the explanations should make correct label predictions in

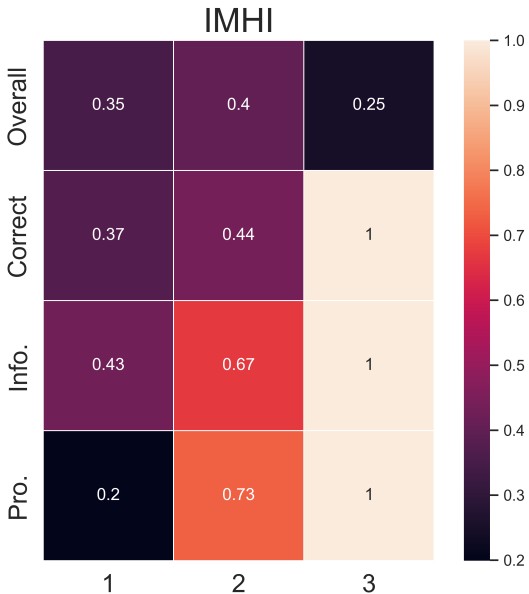

Figure 5: Heatmap of the objective distributions on the dynamic multi-objective dataset built from IMHI. "Pro." and "Info." denote the "Professional" and "Informative" objectives. Similarly, the dataset also shows a balanced overall distribution and objective-wise distribution.

the corresponding mental health analysis task; (2) **Informativeness**: the response should express clear logic and provide consistent evidence; (3) **Professionalism**: the response should provide evidence with high quality and reliability from the perspective of domain experts. Each aspect is divided into four standards rating from 0 to 3. Higher ratings reflect more satisfactory performance and 3 denotes approaching human performance. Each LLM-generated explanation is assigned a score by 2 domain experts for each corresponding objective, followed by the examination of 1 domain expert. All annotators are PhD students majoring in quantitative psychology.

Annotators will be given generated responses from the 4 policy models and need to score and annotate the responses from the following objectives:

**Correctness.** Correctness measures the trustworthiness of the classification results. Annotators should assess whether the classification result is based on facts, has misinformation, and wrong reasoning according to the given post.

- 0: Completely unreliable information with factual hallucination (e.g. non-existent symptoms).
- 1: Partly reliable information with wrong reasoning based on facts.
- 2: Mostly reliable information with non-critical misinformation or wrong reasoning.
- 3: Completely reliable information.

**Informativeness.** Whether the text builds from sentence to sentence to a coherent body of information and logic about mental health and supports the classification results. Annotators should assess if the generated explanation gives consistent supporting evidence to its classifications and is well-structured.

- 0: Inconsistent with the classification results.
- 1: Consistent with the classification results, but with poor readability and several errors.
- 2: Consistent with the classification results. Mostly coherent and easy to read, with few minor errors.
- 3: Consistent with the classification results. Completely fluent, coherent, and error-free.

**Professionalism.** Professionality measures the rationality of the generated explanations by evaluating the evidence that supports the classification results from the psychology perspective. Annotators

should assess whether the explanation includes the following specified common diagnosis criteria of depression. To ensure the quality of the annotation scheme, we invite our domain experts to develop a list of common symptoms related to depression and sort these symptoms by criticality. The domain experts consult the Patient Health Questionnaire (PHQ-9) on determining the symptoms and sorting these symptoms on their knowledge.

Specifically, the following symptoms are checked (sorted by criticality):

- Suicide ideation: Thoughts that you would be better off dead.
- Self-harm ideation: Thoughts of hurting yourself in some way.
- Feeling down, depressed, or hopeless.
- Self-guilt ideation: Feeling bad about yourself — or that you are a failure or have let yourself or your family down.
- **Symptoms above are classified as with high criticality, and symptoms below are classified as with low criticality.**
- Feeling tired or having little energy. Little interest or pleasure in doing things.
- Poor appetite or overeating.
- Trouble falling or staying asleep, or sleeping too much.
- Trouble concentrating on things, such as reading the newspaper or watching television.
- Moving or speaking so slowly that other people could have noticed. Or the opposite — being so fidgety or restless that you have been moving around a lot more than usual
- Uncontrollable sexual desire or sexual frigidity.
- Other symptoms.

Based on the above symptoms, the annotators score the professionality of each explanation with the following criteria:

- 0: The explanation provides no supportive evidence or symptoms with high criticality are missing in the explanation.
- 1: The explanation provides a few supportive evidence, while some symptoms with higher criticality (than provided evidence) are missing.
- 2: The explanation provides several supportive evidence, while some symptoms with lower criticality (than provided evidence) are missing.
- 3: The explanation provides all related supportive evidence in the post.

# E   Sub-task Performance on IMHI Benchmark

We stack *MetaAligner* on different policy models to perform alignment on all 3 objectives: "Correct", "Informative", and "Professional". We include MentaLLaMA-(7B, 13B, 33B) [34], the first open-source instruction-following LLM series for interpretable mental health analysis into the policy models. Details about the 9 sub-tasks are provided in Table 6. The overall performance of *MetaAligner* on the IMHI benchmark and its separation into 9 different sub-tasks are shown in Table 7.

According to the results, the *MetaAligner* models achieve substantial improvement in overall performance on all 11 policy models, with an average of 26.89% advantage on win rates for *MetaAligner*-1.1B, 28.01% for *MetaAligner*-7B, and 36.6% for *MetaAligner*-13B. These results show the general effectiveness of one *MetaAligner* on various upstream models and the feasibility of plug-and-play multi-objective alignment. *MetaAligner* also greatly improves performance on each sub-task. For example, *MetaAligner*-7B outperforms the unaligned outputs by over 25% on 7 sub-tasks. These results indicate that *MetaAligner* alignment can be effectively adapted to tasks that require different knowledge and response formats.

From the policy model scale perspective, *MetaAligner* provides successful alignments to models with sizes ranging from 2B to 70B, significantly extending the size of *MetaAligner* itself. In the extreme case, *MetaAligner*-1.1B advances the win-rate of LLaMA2-Chat-70B outputs by 21.18%, a

Table 5: Details for *MetaAligner* training and datasets. 'preference source' denotes how the preference annotations were obtained.

| Training Information | |
| --- | --- |
| Base Library | Huggingface Transformers |
| Fine-tuning Platform | FastChat |
| GPU Hardware | 4× NVIDIA Tesla A100 80GB GPUs |
| CPU Hardware | 8× Intel(R) Xeon(R) Gold 6342 CPU cores per GPU |
| Hardware Speedup | Flash Attention 2 [8] |
| Quantization for training | BF16 |
| Fine-tuning Strategy | Full fine-tuning |
| Optimizer | Adam |
| Training Epochs | 2 |
| Batch sizes | HH-RLHF: 512 / UltraFeedback: 512 / IMHI: 128 |
| Max token for training | *MetaAligner*-(1.1B, 7B, 13B): 2048/4096/4096 |
| Learning rate | 1e-5 |
| Warm-up ratio | 0.05 |
| Base Model-1.1B | TinyLLaMA-1.1B |
| Base Model-7B/13B | LLaMA2-Chat-(7B, 13B) |

| Dataset Information | |
| --- | --- |
| Dataset Name | **HH-RLHF** |
| License | MIT |
| Train/Val/Test ($\mathcal{D}_p$) | 262,719/15,000/15,000 |
| Train/Val ($\mathcal{D}_e$) | 16,502/1,797 |
| Harmless preference source | Ray2333/gpt2-large-harmless-reward_model |
| License | MIT |
| Helpful preference source | Ray2333/gpt2-large-helpful-reward_model |
| License | MIT |
| Humor preference source | mohameddhiab/humor-no-humor |
| License | Apache-2.0 |
| Test evaluator | GPT-4 |
| Dataset Name | **UltraFeedback** |
| License | MIT |
| Train/Val/Test ($\mathcal{D}_p$) | 252,934/15,000/15,000 |
| Train/Val ($\mathcal{D}_e$) | 82,023/5,000 |
| Instruction_following preference source | GPT-4 |
| Honest preference source | GPT-4 |
| Truthful preference source | GPT-4 |
| Helpful preference source | GPT-4 |
| Test evaluator | GPT-4 |
| Dataset Name | **IMHI** |
| License | MIT |
| Train/Val/Test ($\mathcal{D}_p$) | 5,304/1,051/2,400 |
| Train/Val ($\mathcal{D}_e$) | 3,374/689 |
| Instruction_following preference source | Human annotation |
| Correct preference source | Human annotation |
| Informative preference source | Human annotation |
| Professional preference source | Human annotation |
| Test evaluator | GPT-4 |

| Policy Models | |
| --- | --- |
| LLaMA2-Chat-(7B, 13B, 70B) | https://huggingface.co/meta-llama |
| Gemma-instruct-(2B, 7B) | https://huggingface.co/google |
| Vicuna-(7B, 13B, 33B) | https://huggingface.co/lmsys |
| GPT-3.5-Turbo | https://openai.com/blog/chatgpt |
| Claude-3-Sonnet | https://www.anthropic.com/news/claude-3-family |

---

**Algorithm 2** GPT-4 win-rate computation

---

**Require:** The aligned test dataset: $\{q_i, g_i, R_i^{origin}, R_i^{aligned}\}_{i=1}^n$; Text description for target objective: $\langle r_t \rangle$; Prompting template: $\mathcal{E}(q, r_1, r_2, \langle r \rangle)$

**Ensure:** Win rate of the aligned responses $\omega$.

1:   $W_{origin} \leftarrow \varnothing; W_{aligned} \leftarrow \varnothing$             $\triangleright$ Initialize the judgement set W.

2:   $P_{origin} \leftarrow \varnothing; P_{aligned} \leftarrow \varnothing$       $\triangleright$ Initialize the set P to record the position of the responses.

3:   $win_{origin} = 0; win_{aligned} = 0$          $\triangleright$ Initialize the counter for wining samples.

4:   **for** $i \in \{1, ..., n\}$ **do**

5:      $r_1^{origin}, r_2^{origin}, p_i^{origin} = random\_shuffle(g_i, R_i^{origin})$      $\triangleright$ Random shuffle the origin and ground-truth response. $p_i^{origin}$ denotes the position of $R_i^{origin}$.

6:      $P_{origin} \leftarrow \mathcal{E}(q_i, r_1^{origin}, r_2^{origin}, \langle r_t \rangle)$      $\triangleright$ Prompt for comparing origin and ground-truth response.

7:      $J_i^{origin} \leftarrow Call\text{-}GPT\text{-}4(P_{origin})$          $\triangleright$ Call GPT-4 API to perform judgement.

8:      $W_{origin} \leftarrow W_{origin} \cup \{J_i^{origin}\}$

9:      $P_{origin} \leftarrow P_{origin} \cup \{p_i^{origin}\}$

10:     **if** $J_i^{origin} = p_i^{origin}$ **then**

11:        $win_{origin} = win_{origin} + 1$

12:     **end if**

13:     $r_1^{aligned}, r_2^{aligned}, p_i^{aligned} = random\_shuffle(g_i, R_i^{aligned})$      $\triangleright$ Similar actions for aligned response.

14:     $P_{aligned} \leftarrow \mathcal{E}(q_i, r_1^{aligned}, r_2^{aligned}, \langle r_t \rangle)$

15:     $J_{aligned} \leftarrow Call\text{-}GPT\text{-}4(P_{aligned})$

16:     $W_{aligned} \leftarrow W_{aligned} \cup \{J_i^{aligned}\}$

17:     $P_{aligned} \leftarrow P_{aligned} \cup \{p_i^{aligned}\}$

18:     **if** $J_i^{aligned} = p_i^{aligned}$ **then**

19:        $win_{aligned} = win_{aligned} + 1$

20:     **end if**

21: **end for**

22: $\omega_{origin} = \frac{win_{origin}}{len(W_{origin})}$ $\triangleright$ Calculate win rates for original responses over ground-truth responses.

23: $\omega_{aligned} = \frac{win_{aligned}}{len(W_{aligned})}$ $\triangleright$ Calculate win rates for aligned responses over ground-truth responses.

24: $\omega = \omega_{aligned} - \omega_{origin}$

---

Table 6: Details about the 9 sub-tasks in the IMHI dataset. "Annotation" denotes the reliability of the annotations in the raw data.

| Data | Task | Source | Annotation | Labels/Aspects |
|------|------|--------|-----------|----------------|
| DR | depression detection | Reddit | weak supervision | Yes, No |
| Dreaddit | stress detection | Reddit | human annotation | Yes, No |
| SWMH | mental disorders detection | Reddit | weak supervision | Suicide, Anxiety, Bipolar disorder, Depression, None |
| T-SID | mental disorders detection | Twitter | weak supervision | None, Suicide, Depression, PTSD |
| SAD | stress cause detection | SMS | human annotation | School, Finance, Family, Social Relation, Work, Health, Emotion, Decision, Others |
| CAMS | depression/suicide cause detection | Reddit | human annotation | Bias, Jobs, Medication, Relationship, Alienation, None |
| loneliness | loneliness detection | Reddit | human annotation | Yes, No |
| MultiWD | Wellness dimensions detection | Reddit | human annotation | Spiritual, Physical, Intellectual, Social, Vocational, Emotional |
| IRF | interpersonal risk factors detection | Reddit | human annotation | Thwarted Belongingness, Perceived Burdensomeness |

Table 7: Performance of *MetaAligner*-(1.1B, 7B, 13B) on each IMHI sub-task over different policy models. The GPT-4 judge considers 3 objectives: "Correct", "Informative", and "Professional". The figures show the advantage of aligned outputs over the policy model outputs on *win rate*. Best values for each *MetaAligner* model are highlighted in bold.

| MetaAligner | Policy Model | CAMS | DR | Dreaddit | IRF | loneliness | MultiWD | SAD | SWMH | T-SID | Overall |
|---|---|---|---|---|---|---|---|---|---|---|---|
| **1.1B** | LLaMA2-Chat-7B | -3.4% | +28.0% | +12.67% | +45.67% | +38.0% | +34.67% | +23.33% | +33.0% | +31.33% | +27.04% |
| | LLaMA2-Chat-13B | -25.0% | +27.67% | +22.67% | **+52.33%** | +32.0% | +31.33% | +23.33% | +23.67% | +36.67% | +29.7% |
| | LLaMA2-Chat-70B | +5.33% | +35.0% | +19.0% | +46.33% | +42.0% | -3.0% | +8.0% | +0.33% | +3.0% | +21.18% |
| | Gemma-instruct-2B | **+32.0%** | +4.33% | +37.0% | +40.0% | +34.0% | **+61.0%** | +45.0% | +52.0% | +27.33% | +38.77% |
| | Gemma-instruct-7B | +16.33% | **+51.33%** | **+42.67%** | +44.67% | **+51.0%** | +55.33% | **+49.33%** | **+40.0%** | **+53.67%** | **+44.92%** |
| | MentalLLaMA-7B | +13.33% | +22.67% | +23.33% | +47.33% | +39.67% | +39.33% | +33.33% | +30.67% | +41.0% | +32.29% |
| | MentalLLaMA-13B | +21.34% | +31.0% | +39.0% | +29.67% | +42.33% | +47.66% | +20.67% | +34.33% | +33.0% | +36.03% |
| | MentalLLaMA-33B | -5.44% | -0.33% | -5.33% | +30.67% | +3.0% | +14.0% | +24.33% | +7.0% | +2.33% | +6.66% |
| | Vicuna-7B | +6.0% | -9.33% | +24.0% | +70.67% | +17.67% | +33.33% | +17.33% | +22.0% | +42.0% | +24.85% |
| | Vicuna-13B | +6.0% | -9.0% | +13.67% | +71.67% | +20.33% | +32.67% | +21.0% | +2.0% | +16.67% | +24.97% |
| | Vicuna-33B | +29.33% | +2.0% | -1.0% | +41.0% | +10.0% | +9.67% | +1.33% | -9.67% | +1.33% | +9.33% |
| **7B** | LLaMA2-Chat-7B | +7.67% | +47.0% | +13.0% | +33.0% | +38.33% | +31.0% | +17.0% | +38.67% | **+49.33%** | +30.55% |
| | LLaMA2-Chat-13B | -12.34% | **+51.67%** | +18.0% | +37.67% | **+43.33%** | +29.33% | +22.33% | +18.0% | +38.34% | +27.37% |
| | LLaMA2-Chat-70B | +15.66% | +40.53% | +9.0% | +28.33% | +25.34% | +6.67% | +31.33% | +10.33% | +10.66% | +20.67% |
| | Gemma-instruct-2B | **+45.0%** | +8.33% | +30.0% | +57.0% | +39.0% | **+48.66%** | **+57.0%** | **+52.0%** | +36.33% | **+41.11%** |
| | Gemma-instruct-7B | +32.0% | +42.67% | +20.67% | +45.0% | +36.33% | +43.0% | +46.0% | +40.67% | +20.7% | +38.0% |
| | MentalLLaMA-7B | +20.66% | +50.67% | +19.66% | +35.66% | +30.0% | +27.0% | +40.33% | +36.34% | +41.0% | +35.48% |
| | MentalLLaMA-13B | +25.0% | +45.66% | **+43.66%** | +34.34% | +36.33% | +30.33% | +48.0% | +42.33% | +36.0% | +37.97% |
| | MentalLLaMA-33B | -5.33% | +2.66% | -9.66% | +2.34% | +21.33% | -4.67% | +20.0% | +8.67% | +3.33% | +4.22% |
| | Vicuna-7B | +22.33% | +7.67% | +4.0% | +40.0% | +20.0% | +14.0% | +6.0% | +7.67% | +15.0% | +15.19% |
| | Vicuna-13B | +1.0% | +48.0% | +8.0% | **+67.0%** | +33.0% | +12.0% | +15.0% | +23.0% | +22.0% | +32.33% |
| | Vicuna-33B | -2.0% | +54.0% | +10.0% | +62.0% | +37.0% | +29.0% | +10.0% | +15.0% | +12.0% | +25.22% |
| **13B** | LLaMA2-Chat-7B | +27.33% | +45.0% | +23.33% | +62.67% | +54.33% | +59.34% | +52.0% | +55.33% | **+70.33%** | +49.96% |
| | LLaMA2-Chat-13B | +5.34% | +47.0% | +22.67% | +65.33% | **+56.33%** | +52.33% | +31.33% | +32.0% | +56.67% | +35.55% |
| | LLaMA2-Chat-70B | +28.0% | **+59.0%** | +23.33% | +57.33% | +53.0% | +7.0% | +27.33% | +23.0% | +23.67% | +31.18% |
| | Gemma-instruct-2B | **+52.66%** | +19.66% | **+41.33%** | +60.67% | +50.67% | **+79.0%** | +53.67% | **+63.33%** | +45% | **+51.74%** |
| | Gemma-instruct-7B | +35.33% | +52.0% | +39.0% | +55.34% | +50.0% | +64.0% | **+56.33%** | +49.33% | +61.0% | +50.36% |
| | MentalLLaMA-7B | +40.67% | +38.34% | +29.33% | **+65.67%** | +38.34% | +57.0% | +46.66% | +49.0% | +61.67% | +47.4% |
| | MentalLLaMA-13B | +34.0% | +27.0% | +39.0% | +54.33% | +35.33% | +49.33% | +41.67% | +46.67% | +45.66% | +43.62% |
| | MentalLLaMA-33B | +2.67% | -8.0% | +2.67% | +30.67% | -8.0% | +23.67% | +29.33% | +28.67% | +20.0% | +11.2% |
| | Vicuna-7B | +23.33% | +17.67% | +8.0% | +63.67% | +35.67% | +36.33% | +12.33% | +18.0% | +37.0% | +28.07% |
| | Vicuna-13B | +15.0% | -7.33% | +2.0% | +67.67% | +24.33% | +40.67% | +19.0% | +4.0% | +17.67% | +20.29% |
| | Vicuna-33B | -6.0% | +54.0% | +36.0% | +79.0% | +38.0% | +29.0% | +24.0% | +21.0% | +24.0% | +33.22% |

policy model with $63\times$ more parameters. These results prove *MetaAligner* as a parameter-efficient alignment strategy compared to previous multi-objective alignment methods, where the policy model weights are updated, leading to an inevitable surge of computation resources as policy model sizes grow. Besides the general-domain foundation models, *MetaAligner* also improves the performance by an average of 28.32% on MentalLLaMA models, which are fine-tuned on mental health analysis tasks. These results show that *MetaAligner* can make reasonable corrections on weak responses while maintaining their expertise from domain-specific policy models.

## F  Objective-wise Alignment

### F.1  Experimental Settings

We randomly sample 1,200 queries from the UltraFeedback test set and probe the target policy models to provide responses to all queries, which are regarded as unaligned responses. Then *MetaAligner* is used to align these responses under different objectives, including aligning on each objective (Instruction following, Helpfulness, Honesty, and Truthfulness), and another full alignment on all 4 objectives. During evaluation, GPT-4 is leveraged as a reward model to score these responses considering different objectives. Specifically, the following prompt is developed to obtain the scores:

> $[\mathcal{S}(q, a, \langle r \rangle)]$
> You are a skilled evaluator of helpful AI assistants. You will be presented with one query and a response to this query.
> QUERY: {q} |
> RESPONSE: {a}
> Assign a score ranging from 0 to 10 to this response considering the following aspect: $\{\langle r \rangle\}$. The rating improves as the score rises, where 0 denotes inferior performance and 10 denotes approaching-human performance.
> You must only return the assigned score.

where $q$ and $a$ denotes the target query and response, and $r \in \{$Instruction following, Helpful, Honest, Truthful$\}$ is the target objective for evaluation.

## F.2 Experimental Results

More experimental results are presented in Figure 6. The results are the performance of *MetaAligner*-7B on Gemma-instruct-7B, LLaMA2-Chat-70B, and GPT-3.5-Turbo outputs from the UltraFeedback test set. According to the results, *MetaAligner* can perform effective objective-wise alignment on outputs of different policy models, from the small Gemma-7B model to the competitive commercial models GPT-3.5-Turbo. Unlike full alignment which shows weaker performance as the capability of the policy model increases, objective-wise alignment provides stable improvement for different policy models. For example, aligning "Instruction following" leads to over 20% of approaching-human responses for all policy models, and aligning "Honesty" leads to over 30% of approaching-human responses for all policy models. The reason is that single-objective alignment doesn't involve the complex interactions between multiple objectives, which facilitates the full realization of *MetaAligner*'s potential to improve the response on the corresponding objective.

# G   Implementation Details of Baseline Models

## G.1   SFT-based Methods

Existing SFT-based methods [10, 35] for multi-objective preference alignment explicitly incorporate reward values into the query via prompt engineering, which includes a text marker for each alignment objective and their corresponding value numbers in the current response. The model is trained to predict the response given the enhanced query, which enables it to learn a mapping between the reward value and its reflection in the response. During inference, we achieve alignment by assigning a higher reward value to the target objectives. Specifically, we define the following prompting template for HH-RLHF dataset:

> $[\mathcal{F}(q, r_1, r_2, r_3]$
> <Harmlessness>: $\{r_1\}$; <Helpfulness>: $\{r_2\}$; <Humour>: $\{r_3\}$ | {q}

where $q$ denotes the query, $r_1$, $r_2$, $r_3$ denote the corresponding reward values for the current response, ranging from 1 to 5. We define the following prompting template for UltraFeedback dataset:

> $[\mathcal{F}(q, r_1, r_2, r_3, r_4]$
> <instruction_following>: $\{r_1\}$; <honesty>: $\{r_2\}$; <honesty>: $\{r_3\}$; <helpfulness>: $\{r_4\}$ | {q}

where $q$ denotes the query, $r_1$, $r_2$, $r_3$, $r_4$ denote the corresponding reward values for the current response, obtained from existing reward models.

During inference, we aim to simultaneously align all objectives with 1 model to enable fair comparisons to *MetaAligner*. For UltraFeedback, since all rewards range from 1 to 5, we set all values to 5 during inference: $r_1 = r_2 = r_3 = r_4 = 5$. For HH-RLHF, there are no unified ranges for each objective as all training values are obtained from reward models. Therefore, we set each objective value to its maximum in the training dataset to enable higher alignment. The values are set to: $r_1 = 4.19$, $r_2 = 3.03$, $r_1 = 1.58$.

In calculating the GPU hours, we sum the training hours for fine-tuning the LLaMA-Chat-7B policy model for HH-RLHF and UltraFeedback datasets.

## G.2   MODPO

We implement MODPO by implementing the Controllable Direct Preference Optimization (CDPO) [10] algorithm based on its paper introduction. We bypass the controllable preference SFT stage by utilizing the trained model from SFT-based methods as the foundation model. In the CDPO stage, each query $q$ is accompanied by two responses $c_1$ and $c_2$, where each response is constructed into contrastive pairs. In HH-RLHF, we have the following prompting template:

> $[\mathcal{F}(q, c_1, r_1^1, r_1^2, r_1^3]$
> <Harmlessness>: $\{r_1^1\}$; <Helpfulness>: $\{r_1^2\}$; <Humour>: $\{r_1^3\}$ | {q} | {$c_1$}

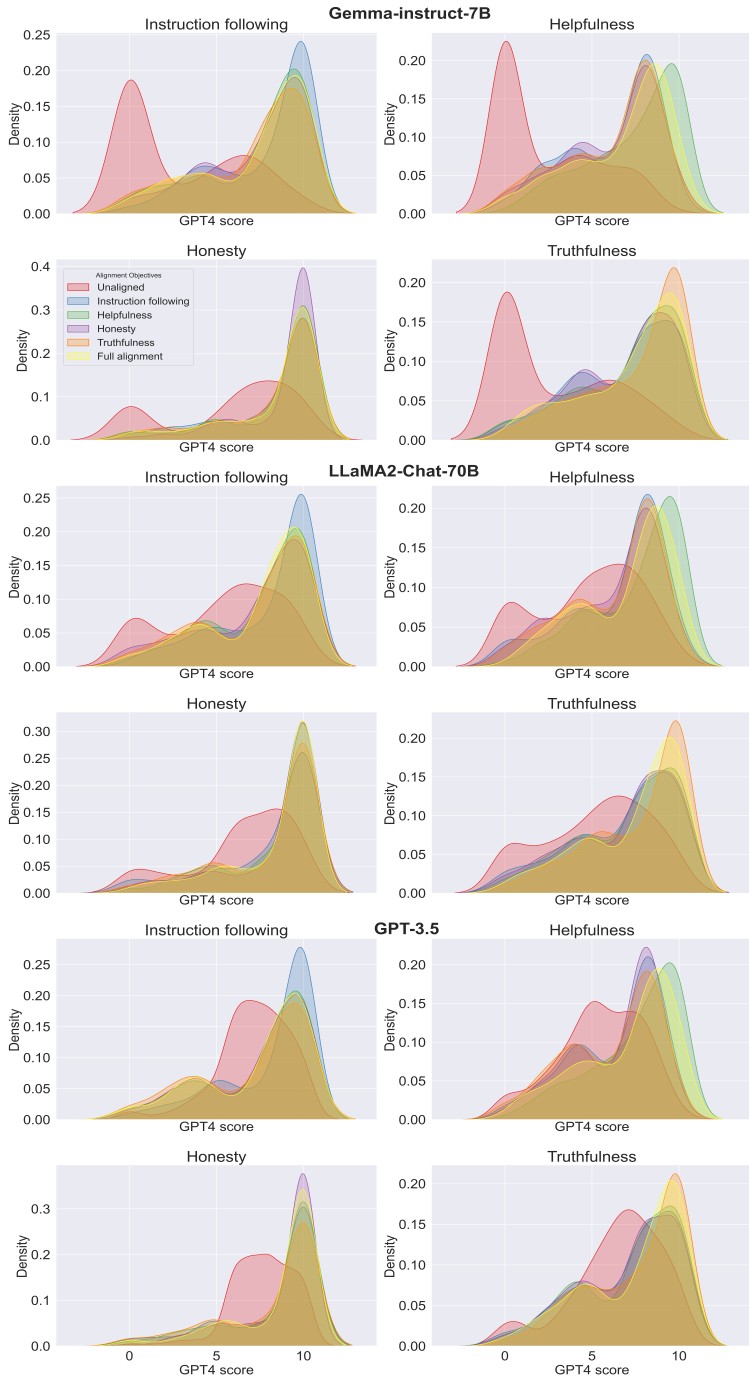

Figure 6: Objective-wise kernel density estimates of GPT-4 evaluation scores under different alignment objectives.

> $[\mathcal{F}(q, c_2, r_2^1, r_2^2, r_2^3]$
> \<Harmlessness\>: $\{r_2^1\}$; \<Helpfulness\>: $\{r_2^2\}$; \<Humour\>: $\{r_2^3\}$ | $\{q\}$ | $\{c_2\}$

where $r_i^j$ denotes the reward value for the $i$-th response on the $j$-th objective. For HH-RLHF, there are no unified ranges for each objective as all training values are obtained from reward models. Therefore, we set each preference value to its maximum in the training dataset to enable higher alignment. The values are set to: $r^1 = 4.19$, $r^2 = 3.03$, $r^3 = 1.58$. Similarly, on UltraFeedback we have:

> $[\mathcal{F}(q, c_1, r_1^1, r_1^2, r_1^3, r_1^4]$
> \<instruction_following\>: $\{r_1^1\}$; \<honesty\>: $\{r_1^2\}$; \<honesty\>: $\{r_1^3\}$; \<helpfulness\>: $\{r_1^4\}$ | $\{q\}$ | $\{c_1\}$

> $[\mathcal{F}(q, c_2, r_2^1, r_2^2, r_2^3, r_2^4]$
> \<instruction_following\>: $\{r_2^1\}$; \<honesty\>: $\{r_2^2\}$; \<honesty\>: $\{r_2^3\}$; \<helpfulness\>: $\{r_2^4\}$ | $\{q\}$ | $\{c_2\}$

For UltraFeedback, since all rewards range from 1 to 5, we set all values to 5 during the DPO training process: $r^1 = r^2 = r^3 = r^4 = 5$. In determining the preference scores, we use the CDPO learning goal to transform the task into a conditional multi-objective optimization problem. Specifically, the reward value $R_i$ for response $c_i$ is calculated as follows:

$$R_i = \sum_{i=1}^{m} \omega_i g_i \tag{7}$$

where $\omega_i$ represents the weight of the $i$-th objective and $g_i$ is calculated as follows:

$$g_i = \begin{cases} -\lambda_i |p_j - r_i^j|, & \text{if } i\text{-th objective is controlled,} \\ r_i^j, & \text{otherwise.} \end{cases} \tag{8}$$

where $\lambda_i$ represents the weight of the controlled objective and $p_j$ is a pre-defined preference value for the $j$-th objective. In our implementation, we set $\omega_i = \lambda_i = 1$. We aim to simultaneously align all objectives with 1 model to enable fair comparisons to *MetaAligner*. For UltraFeedback, since all rewards range from 1 to 5, we set all values to 5 during inference: $p_1 = p_2 = p_3 = p_4 = 5$. For HH-RLHF, there are no unified ranges for each objective as all training values are obtained from reward models. Therefore, we set each preference value to its maximum in the training dataset to enable higher alignment. The values are set to: $p_1 = 4.19$, $p_2 = 3.03$, $p_3 = 1.58$. After calculation, the response with a higher reward value $R_i$ is used as the chosen response, and the other response is used as the rejected response for MODPO training. Specifically, the model is trained via the following loss function:

$$\mathcal{L}_{\text{CDPO}} = -\mathbb{E}_{(x,c,y_w,y_l)\sim\mathcal{D}} \left[ \log \sigma \left( \beta \log \frac{\pi_\theta(y_w \mid c, x)}{\pi_{\text{ref}}(y_w \mid c, x)} - \beta \log \frac{\pi_\theta(y_l \mid c, x)}{\pi_{\text{ref}}(y_l \mid c, x)} \right) \right] \tag{9}$$

where $x$ denotes the query, $y_w, y_l$ denote the chosen and rejected prompts, $c$ denotes the corresponding condition, $\pi_\theta$ and $\pi_{ref}$ denote the target policy model and the reference model. For implementation, we build our MODPO code based on the **OpenRLHF** library.

In calculating the GPU hours, we include the training hours for tuning the SFT-based policy model using the CDPO algorithm for HH-RLHF and UltraFeedback datasets. We also include the training hours for fine-tuning the original LLaMA2-Chat-7B model for fair comparisons.

### G.3 MORLHF

We use the linear scalarization [26, 19] realization of MORLHF with the KL-divergence regularization. Specifically, the model is trained via the following objective function:

$$\underset{\pi_\phi}{argmax} \, \mathbb{E}_{q\sim\mathcal{D}, y\sim\pi_\phi} \left[ \omega^{\text{T}} \mathbf{R}(q, y) - \beta log \frac{\pi_\phi(y|q)}{\pi_{ref}(y|q)} \right] \tag{10}$$

where $\omega = [\omega_1, ..., \omega_N] \, s.t. \sum_{i=1}^{N} \omega_i = 1, \omega_i \geq 0$ is the heuristic target preference vector, $q, y$ denote the query and the response, $\mathbf{R}$ denotes the reward functions for the target objectives. In implementing the MORLHF algorithm, we first train a reward model for each objective based on random samples from 50% of the training data. Following the reward models we used in building the HH-RLHF dataset, we select the **GPT2-large** as foundation models for the reward model, and optimize the reward models using the following pair-wise loss functions:

$$\mathcal{L}_{rm} = -log(\sigma(R_c - R_r - margin)) \tag{11}$$

where $\sigma$ denotes the Sigmoid function, $R_c$ and $R_r$ denote the reward output of the chosen response and the rejected response, and $margin$ denotes the margin loss for the corresponding response pairs when multiple responses are ranked, such as in UltraFeedback. Secondly, we fine-tune the LLaMA2-Chat-7B policy model with the highest-ranked responses from the HH-RLHF and UltraFeedback datasets to obtain sub-optimal starting points for RLHF. The SFT training process is formalized as follows:

$$\underset{\pi_\phi}{argmax} \, \mathbb{E}_{(q,y) \sim \mathcal{D}} \left[ P_{\pi_\phi}(y|q) \right] \tag{12}$$

where $q$ and $y$ denote the query and its corresponding highest-ranked response. Thirdly, following most works in RLHF, we leverage the PPO algorithm [25] to enable parameterized optimization of the policy model. In linear scalarization, we set $\omega_1 = \omega_2 = ... = \omega_N = \frac{1}{N}$. For implementation, we build our MORLHF code based on the **OpenRLHF** library.

In calculating the GPU hours, we include the training hours for all reward models in HH-RLHF and UltraFeedback datasets, with a sum of seven reward model training processes. We also include the training hours for fine-tuning the original LLaMA2-Chat-7B model for reaching the sub-optimal starting points. Finally, the PPO training hours for HH-RLHF and UltraFeedback are included in the GPU hours.

### G.4  Self-Refinement

We include a prompt engineering-based self-refinement method as a baseline method to further demonstrate the effectiveness of MetaAligner. Specifically, this approach involves obtaining an initial response from the policy model and then prompting the same model to further refine its own output. For the second stage, we utilize the same prompting strategies as *MetaAligner*, which is detailed in Appendix C.2. However, this method often requires aligner models with strong in-context learning capabilities, leading to high inference costs due to larger model sizes or expensive commercial models. Therefore, we select LLaMA2-Chat-70B, a strong policy model as the target policy model for self-refinement. As self-refinement does not involve any training procedures, we do not report its GPU hours as other methods.

