# OpenReview forum: "MetaAligner: Towards Generalizable Multi-Objective Alignment of Language Models"
_NeurIPS.cc/2024/Conference — NeurIPS 2024 poster_

### Official Review · Reviewer_vbsw · 2024-07-01

**Soundness:** 3
**Presentation:** 3
**Contribution:** 2
**Rating:** 7
**Confidence:** 4

**Summary:**

Multi-objective alignment of language models is a significant topic for LLM community, while many prior work are either costly-to-compute or policy-dependent, restricting the further depolyment. This work is aimed at providing a retraining-free and policy-agnostic approach for multi-objective alignment, namely *MetaAligner*.

The framework is composed of 3 stages:
- Data reorganization from multi-objective datasets, getting $D_e, D_p$;
- Supervised model training:
   - Warming up on a randomly downsampled dataset of $D_e$;
   - Finetune on $D_e$, as a equal-preference alignment step;
   - Contrastive training on $D_p$.
- Inference with prompting for even unseen objectives.

Experiments are conducted on three commonly-used datasets, and the improvements are reported as prominent. Furthermore, the authors claim that they can achieve generalizable alignment to unseen objectives, through prompting.

**Strengths:**

Originality:
- The whole 3-stage framework of *MetaAligner* is original.

Clarity:
- This paper is well-written and well-organized.
- Most figures and tables are friendly to the readers.

Significance:
- This is the first policy-agnostic and retraining-free multi-objective alignment approach, to the knowledge of the reviewer.
- The experiments are extensive and very solid, including multiple datasets and base models. The author also conducted extensive experiments comparing *Meta-Aligner* with other baselines.

**Weaknesses:**

*Major*
- The whole framework is complicated and a bit engineering. There is no ablation study on each components. For example, it is better to show the performance if discarding the $D_e$ dataset, which is claimed as novel.
- In Table3, the comparison of *MetaAligner* with other baselines is somehow unfair, since *MetaAligner* utilizes at least double parameters of *MORLHF*, *MODPO*, and *RiC* (the 1.1B version performs worse than baselines, so the reviewer is talking about >=7B version).
- The proposed method is not able to cater to different users' taste, because it can only be aligned to multiple objectives while no weights on them are introduced. But many works[1][2][3][4][5] are able to do this, making the contribution less significant.

*Minor*
- this work is a bit marginal:
  - The reorganization process is just preparing the prompting format, which is trivial.
  - The SFT training is a common practice in the community[1][2][3]. The only difference is the design of a detached aligner.
  - The inference stage is simply prompting engineering, lacking guarantees.
- The *MetaAligner* is not a light-weight approach, as the aligner model is still large, making it hard to be deployed.

[1] Rewards-in-Context: Multi-objective Alignment of Foundation Models with Dynamic Preference Adjustment. arXiv preprint arXiv:2402.10207.

[2] Controllable Preference Optimization: Toward Controllable Multi-Objective Alignment. arXiv preprint arXiv:2402.19085.

[3] Arithmetic Control of LLMs for Diverse User Preferences: Directional Preference Alignment with Multi-Objective Rewards. arXiv preprint arXiv:2402.18571.

[4] Decoding-Time Language Model Alignment with Multiple Objectives. arXiv preprint arXiv:2406.18853

[5] Personalized Soups: Personalized Large Language Model Alignment via Post-hoc Parameter Merging. arXiv preprint arXiv:2310.11564.

**Questions:**

- In Figure 3, why would the performance on some unseen objectives drop after adding more objectives? For example, "repeat" and "readable" for *MetaAligner-7B*.
- How much efforts has the author spent on tuning hyper-parameters of *MetaAligner*?
- How does the author set the preference weights for *MORLHF*, *MODPO*, and *RiC*? Is that fair?

**Limitations:**

The limitations are well discussed in Section A.2.

---

> ### Author Rebuttal · Authors · 2024-08-06
>
> Thank you for reviewing our paper and recognizing its strengths. We hope the following responses can help address the concerns.
>
> **Weakness 1**: MetaAligner has a simple methodology. It triggers 2 novel capabilities:
> 1. instance-level alternation of the alignment objectives without re-training;
> 2. generalizable multi-objective alignment to align unseen objectives in a zero-shot manner.
>
> We designed dynamic objectives reformulation (Sec. 3.1) to build training data with various combinations of target objectives which are used to train MetaAligner in a conditional weak-to-strong correction manner (Sec. 3.2). With the dynamic multi-objective capability triggered by the above modules, We further extend it to generalizable inference (Sec. 3.3) with the in-context learning ability of MetaAligner. All modules are centered on achieving the two capabilities.
>
> We didn't show the ablation results because MetaAligner has a simple and intuitive structure, and most components are essential for the models. We agree that showing the ablation results for $D_e$ can be useful evidence for proving its effectiveness. Due to time limits during rebuttal, we will add these results in the future version of our paper.
>
> **Weakness 2**: We admit that the MetaAligner can introduce extra inference costs (we discussed in limitations, appendix A.2). But during training, MetaAligner has fewer trainable parameters and lower costs:
> 1) All policy model weights are fused and only MetaAligner parameters are updated. When using LLaMA2-7B as policy (Table 3),  MetaAligner-1.1B has much fewer trainable parameters than the baseline methods, but with comparable or better performance. MetaAligner-(7B,13B) have comparable or more trainable parameters, but with much better performance.
> 2) MetaAligner models can be trained once and used for all policy models (see Table 2), while the baseline algorithms have to be trained from scratch for each new policy model, as they update policy parameters.
> 3) The efficiency of MetaAligner improves with larger policy models. Note that we used LLaMA2-7B in Table 3 as the policy model due to limited computation resources. With larger policy models (e.g. LLaMA2-70B), the training cost for baseline methods vastly increases, while the training cost for MetaAligner remains constant.
>
> **Weakness 3**: Compared to the user-specified weights for all objectives, MetaAligner’s weight-free specification of desired objectives and zero-shot alignment on unseen objectives are more useful features that no other methods achieve.
> 1) The optimal weight assignment is hard to determine because the Pareto front for multi-objective alignment is usually hard for humans to perceive and quantify[2]. We believe users are more concerned with properly improving certain objective combinations than quantifying their weights by themselves. With MetaAligner, the users can simply identify their desired objectives, and the model can modify the response accordingly with an implicit optimal Pareto front learned during training. We believe our proposed solution is more user-friendly in practical scenarios.
> 2) Compared to previous methods that require users to assign weights to every trained objective, MetaAligner allows users to customize their own combinations of objectives for each instance, which is more flexible.
> 3) MetaAligner allows users to specify new objectives by providing text descriptions, and perform zero-shot alignment on these unseen objectives, a feature no previous methods have achieved.
>
> **Weakness 4**:
> 1) We believe the core of the reorganization process is not the prompt, but identifying target objectives for each chosen-rejected pair, the key to the dynamic multi-objective alignment.
>
> 2) MetaAligner shows 2 key advantages over previous SFT methods: (1) MetaAligner achieves unlimited simultaneous alignment objectives at the instance level, with the dynamic multi-objective alignment training; (2) MetaAligner achieves zero-shot alignment for unseen objectives, with its in-context learning ability.
>
> 3) Prompt-based inference is the optimal way to leverage the in-context learning ability of MetaAligner to achieve generalizable alignment. It's also widely used in previous works[1][2][4]. Following their settings, we provide empirical evaluations on generalizable alignment (Sec. 4.4). Providing theoretical guarantees is hard and beyond the scope of most related works due to the low interpretability of deep models.
>
> **Weakness 5**: MetaAligner can have extra inference costs, but these are reasonable trade-offs for savings during training. Due to word limits, please refer to our response to Weakness 1.
>
> **Question 1**: Enhancements in one objective can affect performance in certain objectives due to their controversial nature, known as "alignment tax"[2]. For example, aligning on "Fair" with MetaAligner-(7B, 13B) benefits its win rates, but harms performance on objectives such as "Readable" and "Factual" compared to when "Fair" is unaligned. This phenomenon is expected and unavoidable. We will include these analyses in the future paper.
>
> **Question 2**: MetaAligner training is simple. We searched learning rates between 5e-6 and 5e-5. All hyperparameters are presented in Table 5 in the appendix.
>
> **Question 3**: We tried to ensure a fair comparison. For MORLHF, we set uniform preference weights, as the weight search process for over 2 objectives can be costly. The weights determination process of MODPO is presented in appendix G.2. RiC doesn’t involve assigning weights, but a preference-to-reward mapping process (appendix G.1).
>
> **References**:
> [1] Rewards-in-Context: Multi-objective Alignment of Foundation Models with Dynamic Preference Adjustment.
>
> [2] Controllable Preference Optimization: Toward Controllable Multi-Objective Alignment.
>
> [3] Arithmetic Control of LLMs for Diverse User Preferences: Directional Preference Alignment with Multi-Objective Rewards.
>
> [4] SALMON: Self-Alignment with Instructable Reward Models.

---

> > ### Comment · Reviewer_vbsw · 2024-08-10
> >
> > The reviewer thanks the authors for the detailed responses to my questions. On reading the responses, most of my concerns have resolved, while I partially disagree with some arguments:
> > - "MetaAligner’s weight-free specification of desired objectives and zero-shot alignment on unseen objectives are more useful features that no other methods achieve." The reviewer thinks flexible control on weights is significant, especially when involving conflicting objectives.
> > - "MetaAligner achieves unlimited simultaneous alignment objectives at the instance level". The number of simultaneously aligned objectives can not be unlimited, since it should be constrained by #data and #training time.
> >
> > I will keep my ratings.
> >
> > *Another question:*
> >
> > Why the authors implemented another version of MODPO, instead of using https://github.com/ZHZisZZ/modpo?

---

> > > ### Author Response · Authors · 2024-08-10
> > > **Response to Reviewer vbsw's Comments**
> > >
> > > The authors thank the reviewer for reading and considering our rebuttal. We are glad that our response addresses most of the reviewer's concerns. Here we provide brief responses to the reviewer's new questions. We hope the reviewer can further consider these points:
> > >
> > > **Response to Concern  1**: We agree that weight control can be useful in scenarios such as conflicting objectives. Though without weight control, MetaAligner archives other more useful features as outlined in our response to weakness 3. It is worth noting that instead of requiring an explicit weighting action, MetaAligner handles conflicting objectives more simply by automatically considering the weight specification process during alignment.
> > >
> > > **Response to Concern 2**: We are sorry for the inaccuracy in our statement in this sentence. As discussed in Sec. 3.3, the authors mean to say "This simple pattern can **theoretically** lead to unlimited simultaneous alignment objectives". We agree that the actual performance is limited by data and training time. But It is worth noting that according to our experimental results in Sec. 4.4, training on 4 objectives already effectively expands alignment performance on up to 10 objectives, which significantly supports the above statement.
> > >
> > > **Response to Question 1**: We thank the reviewer for pointing out the MODPO implementation problem. Please note that based on the authors' knowledge, the implementation of MODPO wasn't fully released until their recent acceptance to ACL 2024, which was after when most of this work was finished. We tried to implement the algorithm but many details were vague in the original paper. Therefore, we selected an easier-implemented algorithm, the CDPO. We agree that comparing with the MODPO algorithm is important, and will include the new results in the future version of our paper.

---

> > > > ### Comment · Reviewer_vbsw · 2024-08-10
> > > >
> > > > Thank you for your clarification!

---

### Official Review · Reviewer_E7pS · 2024-07-11

**Soundness:** 3
**Presentation:** 3
**Contribution:** 3
**Rating:** 6
**Confidence:** 4

**Summary:**

The author proposes Meta-Objective Aligner (MetaAligner), the first policy-agnostic and generalizable method for multi-objective preference alignment.

**Strengths:**

As a lightweight algorithm, this is particularly meaningful as model parameters continue to grow. Conditional weak-to-strong correction extends weak-to-strong correction and weak-to-strong generalization, which are currently significant research topics.

**Weaknesses:**

1. I am not entirely clear about the difference between this paper and Aligner. The author mentions:
> As shown, conditional weak-to-strong correction of MetaAligner extends Aligner [12] to multi-objective alignment scenarios, which are not directly solvable by Aligner itself.

I do not fully understand the distinction between this paper and Aligner. After reading the Aligner paper, I noticed that the authors effectively expand on multi-objective alignment. However, the introduction seems to lack a description of Aligner, making it difficult for readers to understand.

2. I did not understand what the author is trying to convey with Figure 2. Can the author explain the meaning of Figure 2? The elements it aims to express are very complex. Could there be a more streamlined figure, similar to Figure 1 in the Aligner paper? Is the extension to multi-objective alignment at the methodological level or the dataset level?
3. In the experimental section, how did the authors generate an effective correction dataset based on HH-RLHF, UltraFeedback, and IMHI?
4. After reading the Aligner paper, I observed that it conducted experiments on the three dimensions of "Helpful," "Harmless," and "Honest" as described by the authors in line 42. I hope the authors can add comparative experiments with Aligner in the ablation study section. From a reviewer's perspective, the contribution of this paper to LLM alignment is undeniable, but I am particularly concerned about the effective improvement and extension of MetaAligner compared to Aligner.

My questions are all included in the Weaknesses section, with point 4 being my primary concern. I hope the authors can effectively enhance the ablation study section. I believe that algorithms like MetaAligner and Aligner are more efficient compared to RLHF and DPO. I look forward to the authors' response and will consider raising the score further.

**Questions:**

see above.

**Limitations:**

N/A.

---

> ### Author Rebuttal · Authors · 2024-08-06
>
> We sincerely thank the reviewer for reviewing our paper and recognizing the strengths of our work. We also thank the reviewer for willing to consider raising the scores. The following parts contain our point-to-point responses to the weaknesses and questions. We hope they can help address the reviewer's concerns.
>
> **Weakness 1**: We apologize for not including more details about Aligner in the main body due to page limits. Aligner is an SFT-based single-preference alignment method, which trains the model to correct the weak responses to approach the strong responses. The advantages of MetaAligner over Aligner are as follows:
> 1) MetaAligner is the first work to explore dynamic multi-objective alignment which enables instance-level alteration of the alignment objectives without re-training and unlimited simultaneous alignment objectives. In contrast, Aligner can only perform single-preference alignment. To achieve this capability, MetaAligner is trained to approach chosen responses from rejected responses, considering dynamic target objectives. This training paradigm is not explored by Aligner or any other alignment methods.
> 2) MetaAligner is the first work that can perform generalizable multi-objective alignment, which can align unseen objectives in a zero-shot manner, while Aligner or any other previous multi-objective alignment works can only perform alignment on objectives that they were trained on. To achieve this capability, we innovatively leverage the in-context learning ability of MetaAligner to understand the unseen objectives and plan for alignment, allowing the users to flexibly describe their alignment objectives in a natural language format, another key difference from the methodology of Aligner.
> 3) Different from Aligner, We conduct experiments on datasets in both general (HH-RLHF, UltraFeddback) and mental health (IMHI) domains, comprehensively showing the effectiveness of MetaAligner. We also conduct experiments to prove MetaAligner’s generalizable alignment ability to unseen objectives and evaluate its accuracy in objective-wise alignment, which is not covered by Aligner.
>
> We will include more of these explanations in the future version of our paper.
>
> **Weakness 2**: We apologize for the complexity of Figure 2 and will revise it to provide a clearer depiction of MetaAligner. We will discard the MORLHF and MODPO illustrations, and reorganize the MetaAligner pipeline from left to right as follows: (1) dynamic objective reformulation, introducing the building process of the dynamic dataset; (2) conditional weak-to-strong correction; (3) generalizable inference. We will simplify the current complex illustration within each stage. Due to time limits during rebuttal, we will include the modified figure in the future version of our paper.
>
> The extension to multi-objective alignment occurs at both the methodological and dataset levels. Methodologically, we are the first to propose dynamic multi-objective alignment to enable instance-level alternation of the alignment objectives, which is achieved by objective-aware weak-to-strong corrections. We are also the first to perform generalizable multi-objective alignment to align unseen objectives, which is achieved by MetaAligner’s in-context learning ability. At the dataset level, we use Algorithm 1 to reorganize the dataset, training MetaAligner to handle various combinations of alignment objectives, which is crucial for its flexible and zero-shot alignment capability.
>
> **Weakness 3**: Different from Aligner which utilized LLMs to correct the weak answers to generate correction pairs, we leveraged existing response pairs within alignment datasets (e.g. HH-RLHF, UltraFeedback) to supervise weak-to-strong corrections without needing to explicitly generate correction pairs.
>
> Specifically, MetaAligner is trained to generate improved outputs using weak responses as references and strong responses as supervision, guided by the target objectives from the dynamic multi-objective dataset. The target objectives of each response pair are obtained via their preference annotations in the original dataset. More details are described in Section 3.1.
>
> **Weakness 4**: MetaAligner and Aligner can have fundamentally different evaluation processes. Aligner uses a single-preference approach, generating one correction per policy output and evaluating it across "Helpful," "Harmless," and "Honest" dimensions. In contrast, MetaAligner can explicitly incorporate target objectives in the input, allowing it to generate different corrections for the same output based on specified objectives, a key advantage over Aligner. We didn't include the objective-specific evaluation results because as Section 4.5 demonstrates, MetaAligner usually provides better results when controlling on the target objectives than full alignment.
>
> We agree that including comparisons with Aligner in different evaluation settings would further enhance our arguments. We provide our latest results, comparing the performance of MetaAligner-7B and Aligner-7B on the LLaMA2-7B-chat policy model:
>
> |Method| Harmless   | Helpful | Humour|
> | --- |---|---|---|
> | Aligner-7B | 72.0%|79.9%|70.12%|
> | MetaAligner-7B (full alignment)|77.5%|82.0%|83.0%|
> | MetaAligner-7B (objective-specific alignment)|79.1%|84.29%|84.45%|
>
> For UltraFeedback we have:
>
> |Method|IF|Honest|Truthful|Helpful|
> | --- |---|---|---|---|
> |Aligner-7B|52.38%|44.23%|37.19%|39.1%|
> |MetaAligner-7B (full-alignment)|51.33%|48.0%|55.67%|44.33%|
> |MetaAligner-7B (objective-specific alignment)|57.9%|48.6%|57.92%|48.13%|
>
> These results show that though Aligner is comparable to MetaAligner on objectives such as “Helpful” in HH-RLHF, it significantly underperforms MetaAligner in fine-grained objectives such as “Humour” and “Truthful”. These results show the effectiveness of the dynamic multi-objective alignment method. Objective-specific alignment further improves the performance of MetaAligner.
>
> These results will be included in future versions.

---

> ### Author Response · Authors · 2024-08-13
> **Reminder on Rebuttal and Discussions**
>
> Dear Reviewer E7pS,
>
> This is a gentle reminder that we have provided a detailed response to your concerns. Please note that the discussion period is ending soon. We'd appreciate it if you could find time to check the response and further consider your evaluations of our paper. We look forward to our further discussions based on these new clarifications.
>
> Best wishes,
>
> On behalf of the authors of Paper 5185

---

> > ### Comment · Reviewer_E7pS · 2024-08-13
> > **reply to the author**
> >
> > I got COVID-19 and was unable to respond to the author in time. I sincerely apologize.
> >
> > Thank you for your comprehensive response. Most of my concerns have been thoroughly addressed. I choose to keep the score unchanged.

---

> > > ### Author Response · Authors · 2024-08-14
> > >
> > > The authors thank the reviewer for reading and considering our rebuttal. We are glad that our response thoroughly addressed most of the reviewer's concerns.
> > >
> > > We hope you will soon recover from COVID-19!

---

### Official Review · Reviewer_i5Yu · 2024-07-12

**Soundness:** 2
**Presentation:** 3
**Contribution:** 3
**Rating:** 6
**Confidence:** 4

**Summary:**

This work proposes MetaAligner, a plug-and-play multi-objective alignment method that can generalize to unseen objectives.

**Strengths:**

1. This is a well-written paper that clearly expresses its core ideas.
2. MetaAligner is a lightweight alignment method that is easier to tune in multi-objective scenarios, with lower training overhead compared to baselines.
3. MetaAligner is a plug-and-play alignment method that can generalize to various open-source and closed-source models, enabling multi-objective alignment.
4. MetaAligner has generalizability, capable of aligning with zero-shot objectives.

**Weaknesses:**

1. In the experimental section, the aligned answers are compared to the ground truth answers in the dataset to calculate the win rate. This lacks a more direct performance comparison, making the argument for performance advantage insufficiently robust. For example, comparing different baselines by win rate against the ground truth answers in the dataset is inadequate to demonstrate the performance advantage over the baselines.
2. There is a lack of performance comparison with prompt engineering-based aligners. Specifically, it is unclear whether similar effects could be achieved by prompting the chat model to modify the answers within the prompts.

**Questions:**

1. I am interested in understanding the authors' rationale for choosing an indirect comparison with the ground truth and how they view the validity of this choice, especially considering that many models, after being aligned through RLHF, are likely to already exhibit strong performance on the objectives of test set.
2. In the scenario where columns contain exactly two of the three attributes: Harmless, Helpful, and Humorous, which corresponds to the position of the second column in Figure 2 HH-RLHF. Let $( x_1, x_2, x_3 )$ denote the probabilities of having two characteristics, specifically including $Harmless$, $Helpful$, and $Humorous$. According to the figure, $( x_1 = 0.57 )$, $( x_2 = 0.53 )$, and $( x_3 = 0.85 )$. Now, there are exactly three possible combinations of choosing two out of the three characteristics: $(Harmless, Helpful)$, $(Helpful, Humorous)$, and $(Harmless, Humorous)$, and these combinations are mutually exclusive. Let the probabilities of these three events be $( a_1, a_2, a_3 )$ respectively, such that $( a_1 + a_2 + a_3 = 1 )$. The event `contains two characteristics, with one of them being Harmless` can be described as the disjoint union of the events `only contains (Harmless, Helpful)` and `only contains (Harmless, Humorous)`. Therefore, $( x_1 = a_1 + a_3 )$. Similarly, we have $( x_2 = a_1 + a_2 )$ and $( x_3 = a_2 + a_3 )$. Summing both sides of these three equations, we obtain $( 1.95 = 0.57 + 0.53 + 0.85 = x_1 + x_2 + x_3 = 2(a_1 + a_2 + a_3) = 2 \times 1 = 2 )$. I am sincerely inquiring whether my understanding of Figure 2 is mistaken.

**Limitations:**

The article provides a thorough discussion of its limitations in terms of deployment and the number of objectives to be selected.

---

> ### Author Rebuttal · Authors · 2024-08-06
>
> We sincerely thank the reviewer for reviewing our paper, which provides careful assessments and valuable comments. We also thank the reviewer for recognizing the strengths of our work. The following parts contain our point-to-point responses to the weaknesses and questions. We hope they can help address the reviewer's concerns.
>
> **Response to Weakness 1**: We adopted the win rate evaluation method because it is a widely recognized approach in SOTA benchmarks such as  AlpacaEval [1], Arena-Hard [2], and MT-Bench [3]. For instance, Arena-Hard uses win rates against GPT-4 outputs to measure model performance, a practice followed by many recent alignment studies [4][5]. This method is considered both effective and robust.
>
> In our evaluation of MetaAligner, we calculated win rates for each policy model's output against ground truth answers both before alignment: $W_b$ and after alignment $W_a$. The performance of MetaAligner was quantified by the difference in win rates: $W_a-W_b$, as shown in Table 2.
>
> When comparing MetaAligner performance based on the same policy model (Table 3 and Figure 3), we reported win rates after alignment ($W_a$) directly, because all methods used the same policy model ($W_b$ are the same), ensuring a fair comparison. We believe these methods provide a reliable assessment of MetaAligner's performance advantage.
>
> **Response to Weakness 2**: We agree that including prompt engineering-based aligners would further demonstrate MetaAligner’s effectiveness. This approach involves obtaining an initial response and then prompting the same model to refine it. However, this method often requires aligner models with strong in-context learning capabilities, leading to high inference costs due to larger model sizes or expensive commercial models. MetaAligner offers a cost-effective advantage by employing a smaller model  (e.g., MetaAligner-1.1B) for the refinement stage, reducing inference costs while maintaining competitive performance through supervised fine-tuning on the alignment dataset. One MetaAligner model can also be applied to different policy models.
>
> To further address the concern, we take time to add a new prompt-based baseline using LLaMA2-70B-chat with one-time self-refinement. For HH-RLHF, the results are as follows:
>
> |Method| Harmless   | Helpful | Humour|
> | ----------- | ----------- | ----------- | -----------|
> | LLaMA2-70B-chat      |  +6.9%      |   +12.8%  |   +14.91%   |
> | MetaAligner-1.1B      |  +6.58%      |   +7.42%  |   +22.58%   |
> | MetaAligner-7B      |  +16.58%  |   +14.42%  |   +29.08%   |
>
> For UltraFeedback we have:
>
> |Method| IF   | Honest | Truthful |  Helpful   |
> | ----------- | ----------- | ----------- | -----------| ----------- |
> | LLaMA2-70B-chat      |  +18.9%      |   +30.19%  |   +20.0%   |  +14.9%    |
> | MetaAligner-1.1B      |  +6.0%      |   +12.67%  |   +17.33%   |    +16.33%        |
> | MetaAligner-7B      |  +31.0%      |   +27.0%  |   +31.33%   |     +17.0%            |
>
> According to the results, MetaAligner-7B can significantly outperform LLaMA2-70B-chat performance on 6 out of 7 objectives, with only 10% in inference cost. MetaAligner-1.1B can also achieve better or comparable performance on 4 objectives, but with only 1.5% in inference cost. These results prove the significant advantage of MetaAligner over the refinement-based methods. We will include these results in the future version of our paper.
>
> **Response to Question 1**: As noted in our response to Weakness 1, our evaluation method measures performance based on relative differences in win rates before and after alignment, ensuring that the capability of the policy model does not affect evaluation accuracy (see Table 2). While strong models like LLaMA2-Chat-70B and ChatGPT may start with high win rates (high $W_b$), their influence is neutralized by subtracting these rates from the aligned results ($W_a-W_b$), isolating the contribution of MetaAligner. This approach ensures that our assessment reflects the genuine impact of the MetaAligner modules.
>
> **Response to Question 2**: We thank the reviewer for carefully assessing the statistics and pointing out this mistake. We mistyped one of the figures when drawing the heat map with Python codes. We have rechecked and the percentage for “helpful” should be 0.58. We will correct this error in the future version of the paper.
>
> **References**:
>
> [1] Length-controlled alpacaeval: A simple way to debias automatic evaluators. arXiv: 2404.04475
>
> [2] From Crowdsourced Data to High-Quality Benchmarks: Arena-Hard and BenchBuilder Pipeline. arXiv: 2406.11939
>
> [3] Judging llm-as-a-judge with mt-bench and chatbot arena. NIPS 2023.
>
> [4] SimPO: Simple Preference Optimization with a Reference-Free Reward. arXiv: 2405.14734
>
> [5] Low-Redundant Optimization for Large Language Model Alignment. arXiv: 2406.12606

---

> > ### Comment · Reviewer_i5Yu · 2024-08-11
> >
> > The reviewer deeply appreciates the responses provided by the authors, which have addressed most of my concerns.
> > Regarding the issue with data points, I recommend that the authors conduct a more thorough review of the figures and data presented in the paper. As a side note, the Figure 1 is somewhat complex.

---

> ### Author Response · Authors · 2024-08-12
> **Response to Reviewer i5Yu's Comments**
>
> The authors thank the reviewer for considering our response and raising the score. We are glad that our response addresses most of the reviewer's concerns. We will review and modify the figures and data in the future version of our paper.
>
> Regarding Figure 1, we will simplify the figure to clarify the model structure. We refer to our response to Weakness 2 of reviewer E7pS for a detailed outline of the modification plan.

---

### Official Review · Reviewer_XHXG · 2024-07-16

**Soundness:** 2
**Presentation:** 3
**Contribution:** 2
**Rating:** 3
**Confidence:** 5

**Summary:**

This work extends Aligner to multi-objective alignment scenarios. The main contribution is adding a textual description to each sample in the existing preference datasets to indicate the reason for the preference between chosen and rejected samples. The authors found that MetaAligner is more efficient than previously trained multi-objective alignment algorithms. However, this work is incremental, with some unclear details, insufficient evaluation benchmarks, and a lack of baselines.

**Strengths:**

Combining meta-learning concepts with alignment and guiding the model's alignment direction through textual descriptions makes sense to me and could potentially become an important part of RMs in future RLHF works.

**Weaknesses:**

1. The innovation is incremental, essentially applying Aligner to a new domain and altering training data to change the alignment objectives.

2. The most critical detail, which is how the preference descriptions between each chosen and rejected pair in the existing preference datasets are obtained, is unclear.

3. The authors did not adequately present results on objective datasets such as MATH and HumanEval, nor on mainstream subjective evaluation sets like MT-Bench/AlpacaEval.

4. There is a lack of comparative analysis with previous multi-objective alignment methods, such as "SPO: Multi-Dimensional Preference Sequential Alignment With Implicit Reward Modeling."

**Questions:**

1. How are the objectives in Section 3.1 obtained?

---

> ### Author Rebuttal · Authors · 2024-08-06
>
> We sincerely thank the reviewer for reviewing our paper and providing valuable comments. The following parts contain our point-to-point responses to the weaknesses and questions. We hope they can help address the reviewer's concerns.
>
> **Response to Weakness 1**: We'd like to clarify our novelty and contributions compared to Aligner and other alignment works:
> 1) MetaAligner is the first work to explore dynamic multi-objective alignment which enables instance-level alteration of the alignment objectives without re-training and unlimited simultaneous alignment objectives. In contrast, Aligner can only perform single-preference alignment and other previous multi-objective alignment works can only perform alignment on fixed objectives. To achieve this capability, MetaAligner is trained to approach chosen responses from rejected responses, considering dynamic target objectives. This training paradigm is not explored by Aligner or any other alignment methods.
> 2) MetaAligner is the first work that can perform generalizable multi-objective alignment, which can align unseen objectives in a zero-shot manner, while Aligner or any other previous multi-objective alignment works can only perform alignment on objectives that they were trained on. To achieve this capability, we innovatively leverage the in-context learning ability of MetaAligner to understand the unseen objectives and plan for alignment, allowing the users to flexibly describe their alignment objectives in a natural language format, another key difference from the methodology of Aligner.
> 3) Different from Aligner, we conduct experiments on datasets in both general (HH-RLHF, UltraFeddback) and mental health (IMHI) domains, comprehensively showing the effectiveness of MetaAligner. We also conduct experiments to prove MetaAligner’s generalizable alignment ability to unseen objectives and evaluate its accuracy in objective-wise alignment, which is not covered by Aligner. We have provided related codes, models and data in the submitted paper, and they will be released for public usage.
>
> **Response to Weakness 2**: While we did not explicitly use the term “preference descriptions” in our paper, we understand that the reviewer may refer to the dynamic combinations of multi-objective descriptions for each chosen-rejected pair. For each chosen-rejected pair, we first identify the target dynamic objectives via their preference annotations in the original dataset. Then the “preference descriptions” are obtained via a random shuffle of these target objectives, followed by a concatenation of their textual descriptions. More details are illustrated in Sec. 3.1 and Algorithm 1. We also provided examples of the building process in Appendix C.3 to further clarify the process.
>
> **Response to Weakness 3**: We appreciate the opportunity to clarify our choice of datasets:
> 1) Tasks of MATH and HumanEval are not well-suited for evaluating multi-objective aspects, such as "Harmlessness," "Humor," and "Fairness," because they do not engage with the human-centered elements that require subjective judgment and contextual interpretation. Because of these factors, objective datasets are rarely used in multi-objective preference alignment research [1][2][3][4][5];
> 2) While MT-Bench and AlpacaEval are popular for single-preference alignment, they lack support for multi-objective alignment, which is central to our work. The absence of multi-objective statistics on their public leaderboards limits their applicability to our research. Additionally, these benchmarks offer a limited number of testing queries, with only 80 for MT-Bench and approximately 800 for AlpacaEval 2.0;
> 3) We utilized the testing splits from HH-RLHF and UltraFeedback, both of which are widely recognized in multi-objective preference alignment research ([1][5]). These datasets provide extensive testing queries (15,000 each), ensuring a comprehensive evaluation. We also included the IMHI benchmark to assess performance in the mental health analysis domain, with 2,400 testing queries available. All our testing data is publicly accessible, and our methods are reproducible, ensuring the reliability of our results.
>
> **Response to Weakness 4**: While SPO is indeed relevant, it was first submitted on May 21, 2024. Given that the NeurIPS 2024 main conference paper deadline was on May 22, 2024, it was not feasible to incorporate SPO into our study within such a limited timeframe. Our paper includes comparisons with representative multi-objective alignment methods such as MORLHF, CDPO, and RiC, which effectively demonstrate the strengths of our approach. We acknowledge the value of including SPO and plan to add it as a baseline in future work. However, given the timing constraints, we believe the absence of SPO should not be considered a weakness at this stage.
>
> In our work, we selected MORLHF, MODPO, and SFT-based methods as baseline works, which are widely used baseline methods in previous works[1][3][4][5]. We also do not observe a significant decrease in the number of our compared baseline methods compared to these works.
>
> **Response to Question 1**: We guess the reviewer refers to how the objective combinations are formed, please see our response to Weakness 2, where we detail the process of creating dynamic multi-objective combinations.
>
> **References:**
>
> [1] Rewards-in-Context: Multi-objective Alignment of Foundation Models with Dynamic Preference Adjustment. arXiv preprint arXiv:2402.10207.
>
> [2] Controllable Preference Optimization: Toward Controllable Multi-Objective Alignment. arXiv preprint arXiv:2402.19085.
>
> [3] Beyond One-Preference-Fits-All Alignment: Multi-Objective Direct Preference Optimization. arXiv:2310.03708
>
> [4] Arithmetic Control of LLMs for Diverse User Preferences: Directional Preference Alignment with Multi-Objective Rewards. arXiv preprint arXiv:2402.18571.
>
> [5] SPO: Multi-Dimensional Preference Sequential Alignment With Implicit Reward Modeling. arXiv: 2405.12739

---

> > ### Comment · Reviewer_XHXG · 2024-08-13
> >
> > Thank you for your rebuttal, but I plan to keep my score because:
> >
> > Regarding novelty:
> >
> > 1. The authors have not demonstrated that MetaAligner can handle unlimited simultaneous alignment objectives.
> > 2. Generalizing Aligner to multi-objective alignment is an intuitive idea.
> > 3. The use of different experimental datasets cannot be considered a measure of novelty.
> >
> > Regarding the experimental datasets, although the authors argue that benchmarks like MATH, HumanEval, MT-Bench, etc., are not suitable for the multi-objective alignment scenario, please note that the goal of multi-objective alignment is still to achieve alignment. Therefore, ensuring that the basic LLM capabilities do not decline is a fundamental requirement.

---

> ### Author Response · Authors · 2024-08-13
> **Reminder on Rebuttal and Discussions**
>
> Dear Reviewer XHXG,
>
> This is a gentle reminder that we have provided a detailed response to your concerns. Please note that the discussion period is ending soon. We'd appreciate it if you could find time to check the response and further consider your evaluations of our paper. We look forward to our further discussions based on these new clarifications.
>
> Best wishes,
>
> On behalf of the authors of Paper 5185

---

> ### Author Response · Authors · 2024-08-14
> **Response to Reviewer XHXG's Comments**
>
> The authors sincerely thank the reviewer for considering our response. Here we provide brief responses to the reviewer's new comments. We hope the reviewer can further consider these points:
>
> **Response to Comment 1**: As discussed in Sec. 3.3, the authors mentioned "This simple pattern can theoretically lead to unlimited simultaneous alignment objectives". It is worth noting that according to our detailed experimental results in Sec. 4.4, training on 4 objectives already effectively expands alignment performance on up to 10 objectives, which significantly supports the above statement. We believe they are strong evidence of the generalizability of MetaAligner.
>
> **Response to Comment 2**: Please note that though we incorporated the training method of Aligner, all modules are centered on achieving the two capabilities.
> 1) instance-level alternation of the alignment objectives without re-training;
> 2) generalizable multi-objective alignment to align unseen objectives in a zero-shot manner.
>
> We designed dynamic objectives reformulation (Sec. 3.1) to build training data with various combinations of target objectives which are used to train MetaAligner in a conditional weak-to-strong correction manner (Sec. 3.2). With the dynamic multi-objective capability triggered by the above modules, We further extend it to generalizable inference (Sec. 3.3) with the in-context learning ability of MetaAligner.
>
> **Response to Comment 3**: The authors would like to clarify that by mentioning our experiments on the mental health analysis domain, we didn't mean to list it as a novelty, but as an empirical contribution that Aligner or other alignment methods didn't consider.
>
> **Response to Comment 4**: The authors would like to clarify that testing on UltraFeedback and HH-RLHF also thoroughly evaluates the basic LLM capabilities on alignment.
> 1) The testing data covers most categories reflected in benchmarks such as MT-Bench, including Writing, Reasoning, QA, Extraction, etc.
> 2) Our testing data has much more samples than the mentioned benchmarks, which allows a thorough evaluation of the capabilities of the LLMs.

---

### Decision · Program_Chairs · 2024-09-25

**Decision:**

Accept (poster)

**Comment:**

Paper presents MetaAligner, a multi-objective preference alignment of language models. It is both policy-agnostic (working with various LLMs) and generalizable to unseen objectives.

I lean towards a borderline acceptance as a poster presentation: The paper addresses a relevant problem of multi-objective alignment in a way that could be beneficial, a policy-agnostic, retrained-free approach for diverse preferences is a desirable goal in the field.
Reviewers i5Yu, E7pS, and vbsw all see merit in the work, acknowledging its potential and highlighting aspects like novelty, clarity, and experimental effort.

Core ideas are worth exploring further despite some weaknesses.

The authors have actively engaged with reviewers, demonstrating a willingness to clarify misunderstandings and addressed concerns.